# Peripheral natural killer cells in chronic hepatitis B patients display multiple molecular features of T cell exhaustion

Marie Marotel[1], Marine Villard[1,2], Annabelle Drouillard[1], Issam Tout[1], Laurie Besson[1,2], Omran Allatif[1], Marine Pujol[1], Yamila Rocca[1], Michelle Ainouze[1], Guillaume Roblot[1], Sébastien Viel[1,2], Melissa Gomez[3], Veronique Loustaud[3], Sophie Alain[4], David Durantel[5], Thierry Walzer[1]*, Uzma Hasan[1]*, Antoine Marçais[1]*

[1]CIRI, Centre International de Recherche en Infectiologie, Team Innate Immunity in Infectious and Autoimmune Diseases, Univ Lyon, Inserm, Université Claude Bernard Lyon 1, CNRS, Lyon, France; [2]Service d'Immunologie biologique, Hôpital Lyon Sud, Hospices Civils de Lyon, Lyon, France; [3]CHU Limoges, Service d'Hépatogastroentérologie, U1248 INSERM, Université Limoges, Limoges, France; [4]Département de Microbiologie, CHU de Limoges, Faculté de médecine-Université de Limoges, Limoges, France; [5]Centre de Recherche en Cancérologie de Lyon (CRCL), INSERM, U1052, CNRS, Université de Lyon, Lyon, France

**Abstract** Antiviral effectors such as natural killer (NK) cells have impaired functions in chronic hepatitis B (CHB) patients. The molecular mechanism responsible for this dysfunction remains poorly characterised. We show that decreased cytokine production capacity of peripheral NK cells from CHB patients was associated with reduced expression of NKp30 and CD16, and defective mTOR pathway activity. Transcriptome analysis of patients NK cells revealed an enrichment for transcripts expressed in exhausted T cells suggesting that NK cell dysfunction and T cell exhaustion employ common mechanisms. In particular, the transcription factor TOX and several of its targets were over-expressed in NK cells of CHB patients. This signature was predicted to be dependent on the calcium-associated transcription factor NFAT. Stimulation of the calcium-dependent pathway recapitulated features of NK cells from CHB patients. Thus, deregulated calcium signalling could be a central event in both T cell exhaustion and NK cell dysfunction occurring during chronic infections.

*For correspondence:
thierry.walzer@inserm.fr (TW);
uzma.hasan@inserm.fr (UH);
antoine.marcais@inserm.fr (AMç)

**Competing interests:** The authors declare that no competing interests exist.

## Introduction

Hepatitis B virus (HBV) infection results in immune-mediated viral clearance in 90–95% of adults. The remaining 5–10% fail to control viral infection due to failure in type one cellular immunity, thus leading to chronic hepatitis B (CHB) (*Lai et al., 2003*). As a result, more than 250 million individuals worldwide are chronic HBV carriers (*Plummer et al., 2016*). Natural killer (NK) cells are endowed with antiviral properties such as IFN-γ and TNF-α secretion as well as cytotoxicity that could contribute to HBV clearance (*Fisicaro et al., 2019*). In patients, NK cells are activated during acute HBV infection before the onset of adaptive immunity (*Dunn et al., 2009*; *Fisicaro et al., 2009*; *Zhao et al., 2012*; *Lunemann et al., 2014*; *Yu et al., 2018*). However, in the chronic phase of the disease, NK cells present impaired functions, which could contribute to viral persistence. Indeed, NK cells from CHB patients are characterised by a decreased capacity to produce cytokines such as IFN-γ and TNF-α despite maintaining or even increasing their cytotoxic capacity (*Oliviero et al., 2009*;

*Peppa et al., 2010*; *Sun et al., 2012*; *Tjwa et al., 2011*). This phenomenon has been termed 'functional dichotomy' (*Oliviero et al., 2009*). NK cell functions are controlled by the relative strength of positive and negative signals triggered by the ligation of activating or inhibitory receptors (*Lanier, 2005*). In HBV-infected patients, this balance might be skewed as a decrease in the expression of several activating receptors was previously observed in patient's NK cells (*Lunemann et al., 2014*; *Sun et al., 2012*; *Tjwa et al., 2011*; *Heiberg et al., 2015*). Furthermore, NK cells from CHB patients with high viral loads or liver damage display increased expression of certain inhibitory receptors including immune-checkpoint (ICP) markers such as NKG2A or T cell immunoglobulin and mucin domain containing 3 (TIM3) (*Ju et al., 2010*; *Li et al., 2013*). Moreover, cytokines also participate positively or negatively in the control of NK cell functions (*Marçais et al., 2013*; *Marçais et al., 2014*; *Viel et al., 2016*; *Marçais et al., 2017*). Molecularly, we and others have shown that the mechanistic target of rapamycin (mTOR) pathway integrates positive and negative signals derived from cytokines such as IL-15, IL-12, or TGF-β, respectively, to control NK cell metabolism and functions (*Marçais et al., 2014*; *Viel et al., 2016*; *Marçais et al., 2017*; *Donnelly et al., 2014*). In the context of CHB, reports also involved immuno-modulatory cytokines such as IL-10 and TGF-β in the emergence/maintenance of the dysfunctional phenotype (*Peppa et al., 2010*; *Sun et al., 2012*; *Li et al., 2018*). However, despite these pieces of evidence, a molecular framework to explain NK cell dysfunction is still missing. This is in contrast with the situation that prevails in the T cell field where dysfunction is also observed in contexts of persistent stimulation encountered during chronic infection or cancer: a phenomenon termed exhaustion. Indeed, T cell exhaustion has been defined as a stepwise differentiation process arising in situations of chronic stimulation and combining (1) a gradual loss of effector functions and proliferative capacities, (2) an altered metabolism, (3) the expression of defined ICP functioning as inhibitory receptors, and (4) a specific transcriptional and epigenetic program (*McLane et al., 2019*). The transcription factors responsible for the appearance of the exhausted phenotype in T cells have recently been identified and include NFAT, TOX, and NR4A family members (*Martinez et al., 2015*; *Khan et al., 2019*; *Seo et al., 2019*; *Alfei et al., 2019*; *Scott et al., 2019*; *Yao et al., 2019*; *Liu et al., 2019*). Mechanistically, it was proposed that NFAT activation resulted from unbalanced signalling due to defective co-stimulation leading to preferential activation of the signalling branch dependent on $Ca^{2+}$ flux (*Martinez et al., 2015*). NFAT would then behave as an initiating transcription factor by further driving the expression of TOX and NR4As (*Seo et al., 2019*; *Chen et al., 2019*). By contrast, the mechanisms that deteriorate NK cell functions during chronic infections and how they relate to the phenomenon of exhaustion as defined in the T cell field remain weakly defined.

To shed light on these issues, we established a cohort of CHB patients and healthy donors (HD) and validated the dysfunctional state and altered phenotype of circulating NK cells. Using flow cytometry, we showed that basal and IL-15-mediated activation of the AKT/mTOR pathway were blunted in NK cells of CHB patients. However, this inadequacy did not translate into obvious metabolic defects. To identify the molecular mechanisms leading to dysfunction in an unbiased manner, we performed a transcriptome analysis of circulating NK cells of HD and CHB patients. We found that NK cells of CHB patients presented some of the key molecular hallmarks of exhausted T cells, that is over-expression of transcription factors such as TOX or NR4A-family and their ICP targets. Furthermore, we uncovered a transcriptional signature implicating the activation of a partnerless NFAT, another characteristic of exhausted T cells. Mechanistically, this suggested that NK cells were submitted to unbalanced signalling biased towards $Ca^{2+}$-dependent signalling in CHB patients. In order to test whether such unbalance could induce dysfunction, we induced $Ca^{2+}$ flux in isolation in control NK cells. This treatment altered several phenotypic and functional parameters in a manner similar to CHB infection, thus providing molecular insight into the regulation of the dysfunctional state. Altogether, these data distinctly show that circulating NK cells in CHB patients exhibit key molecular features reminiscent of T cell exhaustion, a knowledge that could inform future immunotherapy strategies.

# Results

## NK cell functionality is impaired in CHB patients

We constituted a cohort of CHB patients and HD controls. Clinical parameters and statistics are presented in *Table 1*. The 32 patients constituting our cohort are in the immune inactive phase characterised by persistent HBV infection of the liver, absence of significant necroinflammatory disease (data not shown), low serum HBV DNA levels, and normal serum aminotransferases. HD controls were sex and age matched to CHB patients. Similar to previously described cohorts, a higher proportion of CHB patients than HD were seropositive for Human Cytomegalovirus (HCMV) (*Table 1*, *Figure 1—figure supplement 1A*), a characteristic that led to higher representation of the adaptive NKG2C$^+$ subset in the NK cell population (*Figure 1—figure supplement 1B*). Peripheral blood mononuclear cells (PBMCs) were isolated and NK cells identified as CD56$^+$CD7$^+$/CD3$^-$/19$^-$/14$^-$/4$^-$ by flow cytometry (*Figure 1—figure supplement 2*). NK cell effector capacities were then analysed. Four different effector functions were measured: cytotoxicity and the production of three cytokines (IFN-γ, MIP1-β, and TNF-α). We measured these readouts in response to cell lines activating different receptors: K562 and Granta coated with Rituximab. As shown in *Figure 1A*, NK cells of CHB patients exhibited normal degranulation in response to K562 stimulation. Yet, basal levels of CD107a were increased in NK cells from CHB patients suggestive of recent stimulation. As CD107a exposure is only a surrogate marker of cytotoxicity, we directly measured cytotoxic capacity against K562 cells. For this purpose, we used a K562 sub-clone expressing the NanoLuc enzyme, which, upon cell lysis, is released in the culture medium allowing accurate quantification of cell death (*Hayek et al., 2019*). As shown in *Figure 1B*, the cytotoxic capacity of NK cells was not affected by HBV infection. In contrast, a significantly decreased capacity to produce IFN-γ and MIP1-β upon K562 stimulation was observed (*Figure 1C*). TNF-α secretion capacity was also decreased and tightly correlated with the decrease in IFN-γ secretion capacity (*Figure 1—figure supplement 3A*); however, this decrease did not reach statistical significance. We then stimulated NK cells with Granta cells coated with Rituximab. To our knowledge, this stimulus has not been previously tested in CHB patients. Degranulation was normal in CHB patients as measured by CD107a surface exposure (*Figure 1—figure supplement 3B*), while IFN-γ production was decreased (*Figure 1D*). The production of MIP1-β and TNF-α was comparatively less affected. In parallel, cytokine production was also measured in response to IL-12/18 stimulation (*Figure 1E*). No significant difference in IFN-γ, MIP1-β, or TNF-α production was observed between CHB patients and HD. Overall, our results demonstrate a defect in cytokine secretion by NK cells from CHB patients when stimulated with MHC-I-deficient or antibody-coated targets, but not with IL-12/18.

**Table 1.** Characteristics of patients and HD enrolled in the study.

| Parameter | CHB Patients 1st cohort | Healthy Donors 1st cohort | CHB Patients 2nd cohort | Healthy Donors 2nd cohort |
|---|---|---|---|---|
| Number | 32 | 30 | 12 | 10 |
| Age (y) Mean Median (range) | 39 ± 13 38 (19–77) | 38 ± 13 36.5 (20–65) | 33 ± 13 30.5 (20–57) | 37 ± 8 38 (26–50) |
| Sex, n (%) Male Female | 20 (62.5) 12 (37.5) | 22 (73) 8 (27) | 8 (66.6) 4 (33.3) | 5 (50) 5 (50) |
| HBV load (IU/mL) Median | 2844 (10–62,225) | / | 1919 (18–32,785) | / |
| HBsAg (IU/mL) Median | 5770 (0.05–34,971) | / | 17,306(4,189–29,890) | / |
| ALT (IU/mL) Median | 20 (6–56) | NA | 21 (12–35) | NA |
| AST (IU/mL) Median | 21 (11–34) | NA | NA | NA |
| HCMV seropositive, % | 86 | 38 | NA | NA |

ALT, serum alanine aminotransferase levels; AST, aspartate transaminase levels; NA, not available.

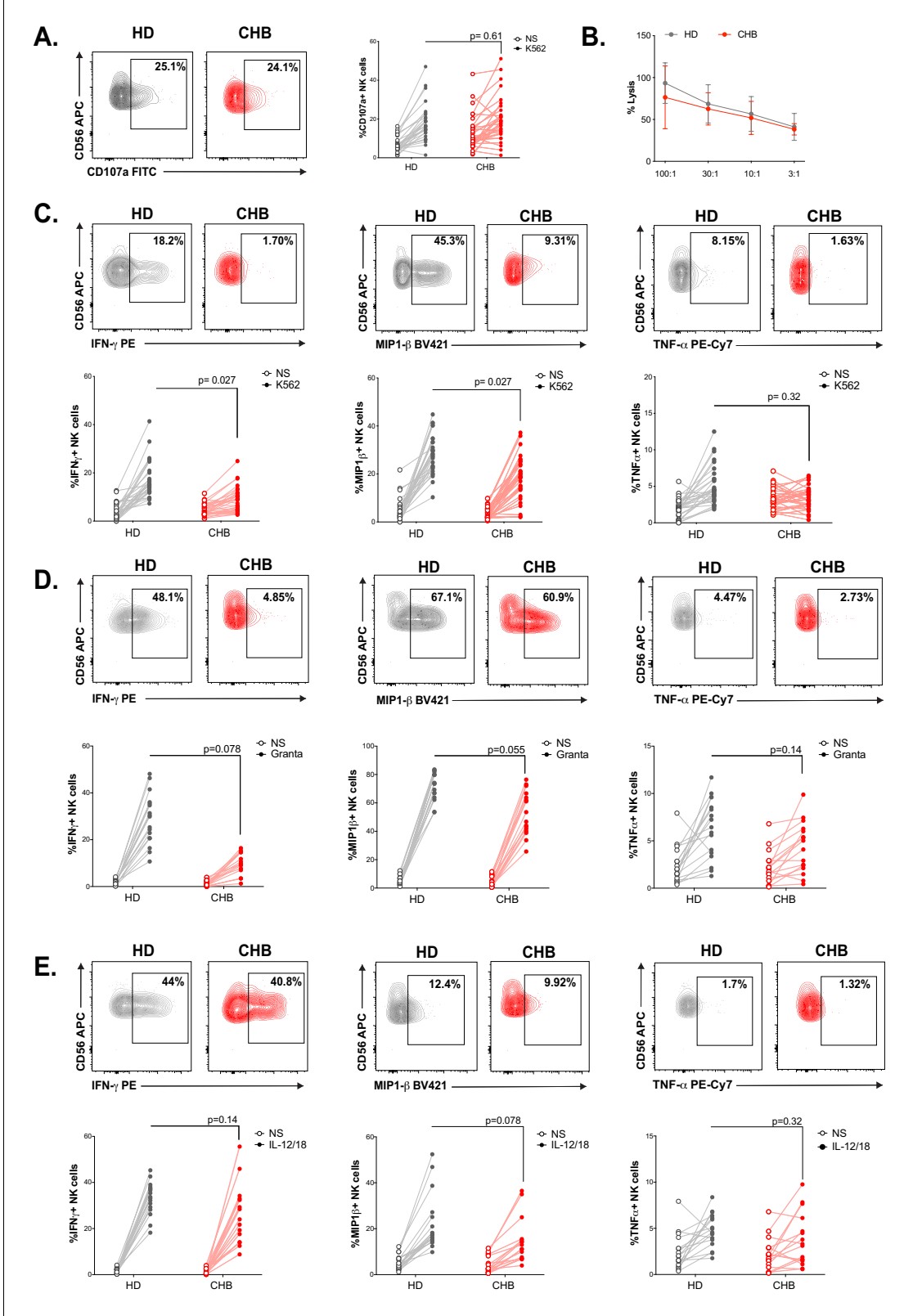

**Figure 1.** NK cell functionality is impaired in CHB patients. (A) PBMCs from HD (n = 30) or CHB patients (n = 32) were co-cultured with K562 during 4 hr, and the proportion of NK cells expressing CD107a was determined by immunostaining. Representative flow-cytometry plots as well as proportion of CD107a[+] NK cells are shown for each individual. (B) PBMCs from HD (n = 17) or CHB patients (n = 15) were co-cultured with K562 NanoLuc at the indicated effectors:targets ratios during 4 hr. Supernatants were then collected to measure bioluminescence. Shown in the figure is the average

*Figure 1 continued on next page*

*Figure 1 continued*

bioluminescence ± SD in an experiment with five HD and five CHB representative of four independent experiments. PBMCs from HD or CHB patients were co-cultured with K562 (**C**), Rituximab-coated Granta (**D**), or with IL-12 and IL-18 (10 ng/mL each) during 4 hr. Intracellular stainings for the indicated cytokines were performed. Representative flow-cytometry plots as well as proportion of NK cells expressing the indicated molecule is shown for each individual (**C**, n = 30 HD and n = 32 CHB, **D** and **E**, n = 17 HD and n = 15 CHB). Statistical analysis was performed by logistic regression as described in the Materials and methods section, and adjusted p-values are indicated on the graph. NS: non-stimulated.

The online version of this article includes the following figure supplement(s) for figure 1:

**Figure supplement 1.** HCMV status.
**Figure supplement 2.** Gating strategy.
**Figure supplement 3.** Functional properties.

## NK cells from CHB patients display an altered phenotype

Next, we measured the percentage of total NK cells as well as the representation of subsets defined by CD56 expression levels following the gating strategy depicted in *Figure 2—figure supplement 1*. We observed that the overall percentage of NK cells was decreased in CHB patients, but this did not impact the relative representation of CD56$^{bright}$ vs CD56$^{dim}$ subsets (*Figure 2A*). In order to determine whether the poor functionality could be explained by altered expression of certain activating or inhibitory receptors, we characterised the phenotype of circulating NK cells in CHB patients compared to HD. The expression of activating receptors CD16 and NKp30 was significantly decreased, while that of NKG2D was increased in CHB patients (*Figure 2B*); the expression of NKp46 and DNAM activating receptors was not statistically different between HD and patients (*Figure 2B*). We also observed an overall reduction in the expression of the inhibitory receptors CD160, KLRG1, and NKG2A (*Figure 2C*). However, this variation was statistically significant only for CD160. Of note, the changes we observed for NKp30, CD16, NKG2D, and CD160 were restricted to the CD56$^{Dim}$ subset (*Figure 2—figure supplement 1A,B*). During NK cell education, the expression of self-engaged inhibitory receptors impacts the content of cytotoxic granules (*Goodridge et al., 2019*). We thus measured the level of the cytolytic proteins Perforin and Granzyme B in NK cells of CHB patients and HD. We did not detect any difference in the expression of these molecules nor in total NK cells neither in the CD56$^{Dim/Bright}$ subsets in accordance with the fact that cytotoxic capacities are preserved (*Figure 2D*, *Figure 2—figure supplement 1C*). Collectively, our results show that dysfunctional NK cells in CHB patients have an altered expression of both activating and inhibitory NK cell receptors but a normal expression of cytotoxic molecules.

## mTOR activation is impaired in NK cells from patients

We have previously shown that NK cell responsiveness is commensurate to the activity of the kinase mTOR (*Marçais et al., 2017*). In addition, the activity of the AKT/mTOR pathway is frequently blunted in exhausted T cells (*McLane et al., 2019*). We thus measured mTOR activation at basal state and in response to IL-15 stimulation in NK cells from CHB patients and HD. mTOR takes part in two distinct complexes: mTORC1 and mTORC2. In order to measure the activity of both complexes, we quantified the phosphorylation level of the ribosomal protein S6 (pS6) as well as the phosphorylation level of the kinase AKT on Ser473 (pAKT) downstream mTORC1 and mTORC2, respectively. In addition, we measured the phosphorylation of STAT5 (pSTAT5) as a control as it is induced by IL-15 stimulation while being independent of mTOR activation (*Marçais et al., 2014*). The gating strategy is presented in *Figure 3—figure supplement 1A*. As depicted in *Figure 3A*, *Figure 3—figure supplement 1B,C*, basal levels of pS6 and pAKT were decreased in total and CD56$^{Dim}$, but not CD56$^{Bright}$ NK cells of CHB patients compared to HD, while pSTAT5 levels were not affected. This demonstrated a negative impact of chronic HBV infection on basal mTOR activity. Of note, when focusing on CHB patients, we found that basal or IL-15-induced levels of pS6 were correlated to IFN-γ production, so that patients with low pS6 also showed low IFN-γ production (*Figure 3—figure supplement 1D* and data not shown). We next tested whether mTOR activity induced by IL-15 stimulation was also impacted. Upon IL-15 stimulation, mTOR activation was lower in total NK cells from CHB patients than in NK cells from HD as shown by the decreased phosphorylation of both S6 and AKT (p-values of 0.11 and 0.038, respectively). Importantly, NK cell responsiveness to IL-15 is developmentally regulated, CD56$^{bright}$ NK cells being the most responsive subset (*Wagner et al., 2017*

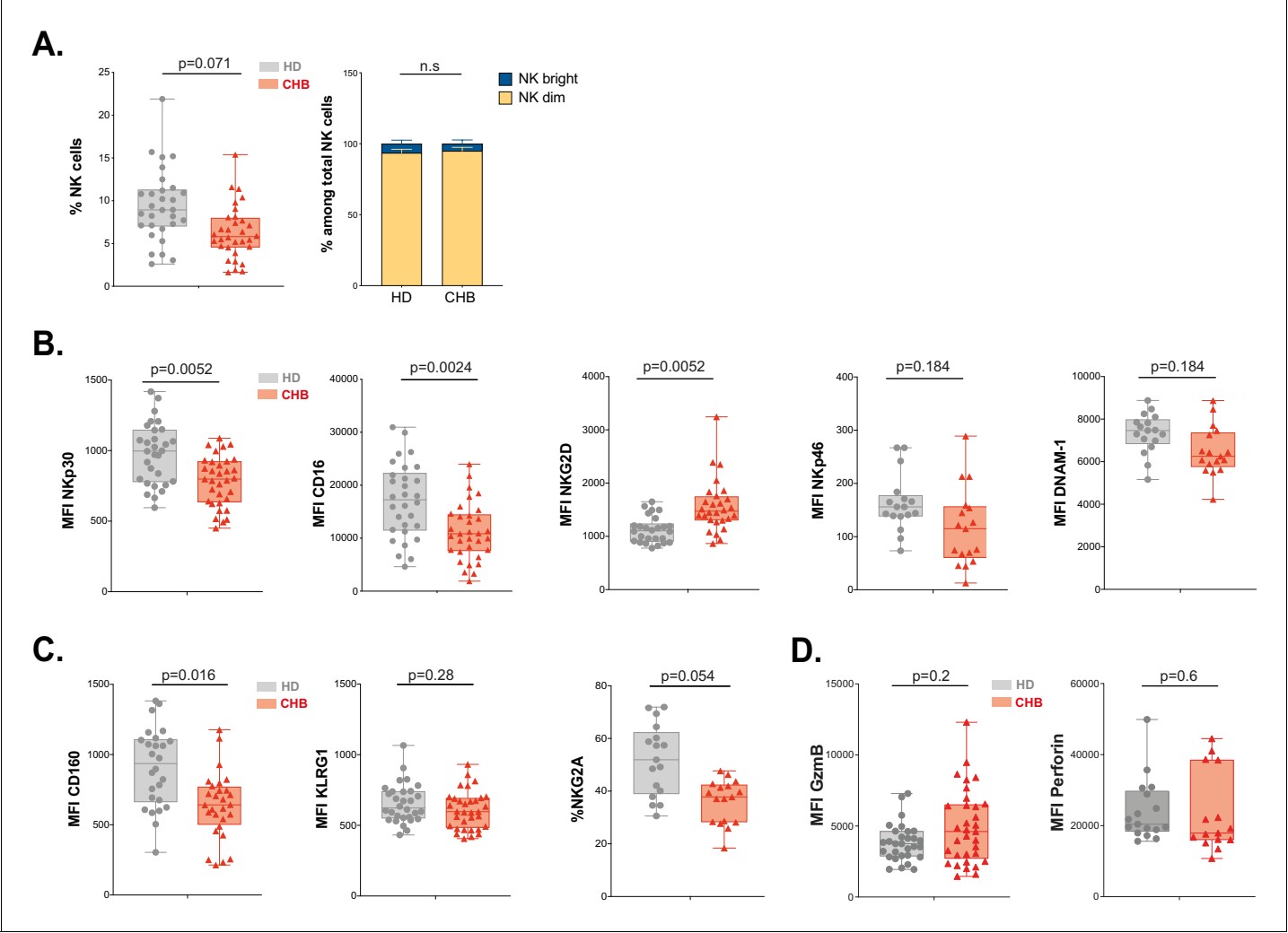

**Figure 2.** NK cell from CHB patients display an altered phenotype. (**A**) PBMCs from 30 HD and 32 CHB samples were stained with live/dead, CD4, CD14, CD19, CD3, CD7, and CD56 antibodies. The percentage of NK cells in each sample (± SD), and the proportion of CD56^bright versus CD56^dim NK cells ± SD), determined by flow cytometry, are shown. (**B,C**) Mean fluorescence intensity (MFI) of indicated NK activating (**B**) and inhibitory (**C**) receptors was determined by flow cytometry on total NK cells among PBMCs from HD or CHB patients. The median expression (± SD) as well as values for each individual are represented, n = 30 HD and 32 CHB samples for NKp30, CD16, NKG2D, KLRG1, and CD160, n = 17 HD and 15 CHB samples for NKp46, DNAM-1, and NKG2A. (**D**) MFI of Granzyme B and Perforin were determined by flow cytometry on total NK cells among PBMCs from HD or CHB patients. The median expression (± SD) as well as values for each individual are represented, n = 30 HD and 32 CHB samples for Granzyme B, n = 17 HD and 15 CHB samples for Perforin. Adjusted p-values are indicated on the graph. n.s: non-significant.

The online version of this article includes the following figure supplement(s) for figure 2:

**Figure supplement 1.** Phenotype of CHB patients NK cells along differentiation.

and our own unpublished data). We thus analysed the impact of chronic HBV infection on mTOR activation in this subset. As shown in *Figure 3—figure supplement 1B*, both phosphorylations were reduced in CHB patients upon IL-15 stimulation. This demonstrated that HBV also negatively impacts mTOR activation induced by IL-15. This defective induction was specific to the mTOR pathway as IL-15 triggering of pSTAT5 was not affected (*Figure 3A*, *Figure 3—figure supplement 1B,C*). We reported that TGF-β is a potent negative regulator of the mTOR pathway in NK cells (*Viel et al., 2016*) and reports indicate higher seric concentration of TGF-β in CHB patient (*Peppa et al., 2010*; *Sun et al., 2012*). We thus measured active TGF-β1 concentration in the serum of HD and CHB patients. Circulating TGF-β1 levels were indeed significantly higher in CHB patients (*Figure 3B*). However, no correlation was observed between TGF-β1 concentration and pS6 level neither at basal

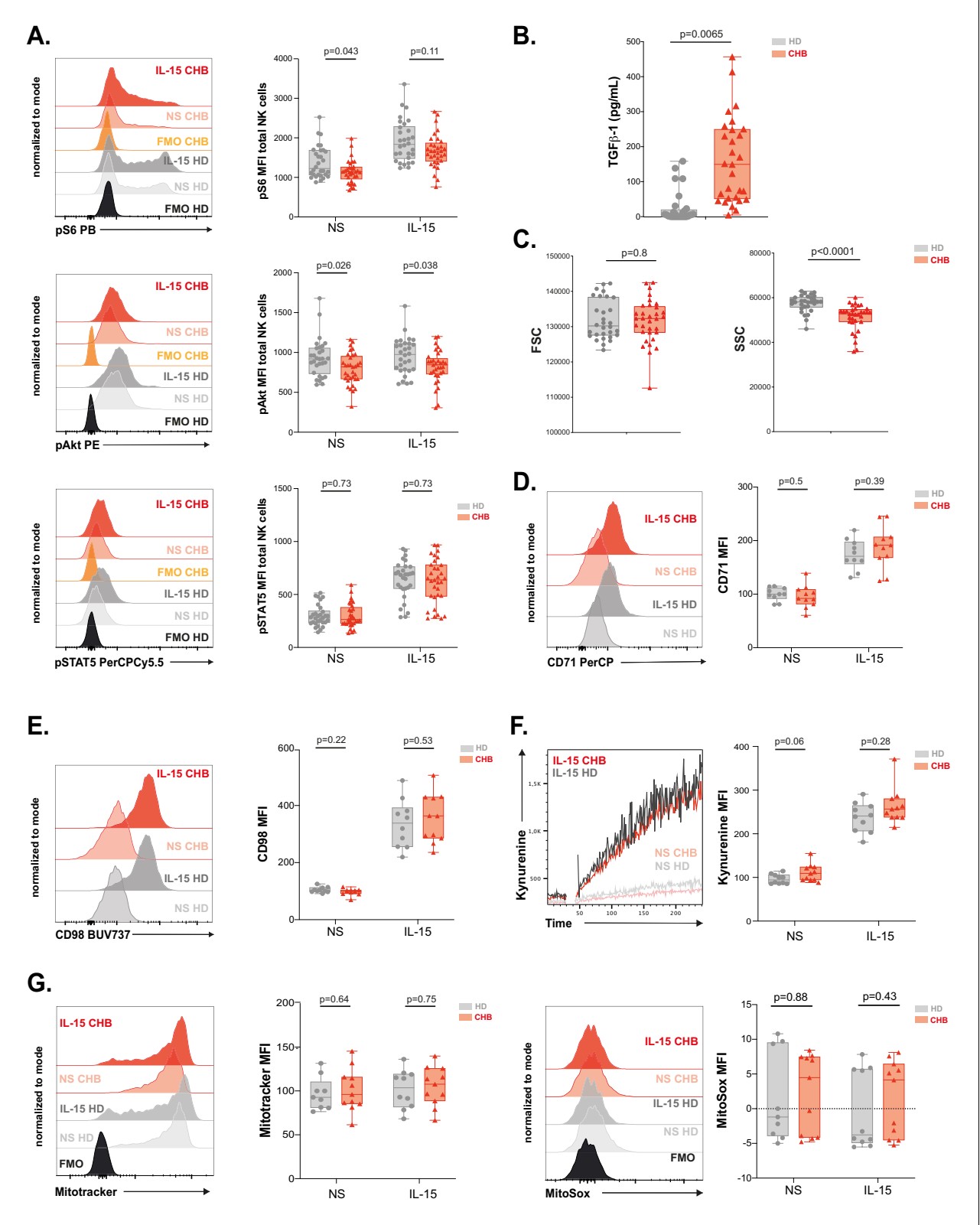

**Figure 3.** mTOR activation is impaired in NK cells from patients. (**A**) PBMCs from HD or CHB patients were stimulated or not with IL-15 at 100 ng/mL for 30 min prior to phospho-epitope staining (pS6 Ser235/236, pAKT S473, and pSTAT5 Y694). Overlays of representative histograms are shown (left panels). The median expression (± SD) as well as MFI values for each individual are represented, n = 30 HD and 32 CHB (right panels). (**B**) Active TGF-β1 levels were measured in serum samples from 30 HD and 32 CHB patients, and the median (± SD) as well as values for each individual are represented. *Figure 3 continued on next page*

*Figure 3 continued*

(**C**) FSC-A and SSC-A parameters on NK cells were measured by flow cytometry in 30 HD and 32 CHB samples. The median of the MFI (± SD) as well as MFI values for each individual are represented. MFI of (**D**) CD71 and (**E**) CD98 was determined by flow cytometry on total NK cells from HD or CHB patients with or without prior stimulation with 100 ng/mL IL-15 O/N. Overlays of representative histograms are shown (left panels). The median expression (± SD) as well as values for each individual are represented, n = 10 HD and 11 CHB samples (right panels). (**F**) PBMCs from 10 HD and 11 CHB patients were stimulated or not O/N with 100 ng/mL IL-15 and their kynurenine uptake capacity was evaluated. A representative kinetic analysis is shown (left panel). MFI was determined in a 30 s time slice centred on 2 min after the beginning of the kynurenine treatment. The mean of the MFI (± SD) as well as MFI values for each individual are represented (right panel). (**G**) PBMCs from 10 HD and 11 CHB patients were stained with Mitotracker Green and MitoSOX Red with or without prior stimulation with 100 ng/mL IL-15 O/N. Overlays of representative histograms are shown (left panels). Median (± SD) as well as MFI values for each individual for the analysed marker are represented (right panels). p-values are indicated on each graph. The online version of this article includes the following figure supplement(s) for figure 3:

**Figure supplement 1.** Gating strategy, phospho-epitopes, and metabolic analysis.

nor after IL-15 stimulation in the CHB patient group (data not shown). Higher TGF-β1 levels could thus participate in the reduction of mTOR activity in CHB patients, but it is likely that other parameters take part in this phenomenon. As mTOR is a key regulator of metabolic networks, we investigated possible metabolic defects in NK cells of CHB patients. For this purpose, we first quantified cellular size and granularity, correlates of metabolic activity. As shown in *Figure 3C*, NK cell size was not affected in CHB patients (forward scatter [FSC] parameter), while they presented decreased granularity (side scatter [SSC] parameter). Furthermore, two nutrient transporters regulated by mTOR (*Marçais et al., 2014*), the heavy chain of system L amino-acid transporter (CD98) and the transferrin transporter (CD71), were also expressed at equivalent levels at basal state and their induction by IL-15 was normal (*Figure 3D,E*). Normal functionality of the system L amino-acid transporter was confirmed in both HD and CHB patients as measured by the uptake of kynurenine (*Sinclair et al., 2018*), a fluorescent derivative of tryptophan carried by the system L transporter (*Figure 3F*). Previous reports have shown that alterations in mitochondrial activity were linked to lymphocyte exhaustion (*Fisicaro et al., 2017*; *Zheng et al., 2019*). Hence, we quantified the global mitochondrial mass using the Mitotracker dye, and mitochondrial production of reactive oxygen species (ROS) using the MitoSOX dye, yet we did not observe any change between the two groups nor at basal state neither after IL-15 stimulation (*Figure 3G*). Of note, the same results were obtained when focusing on the most IL-15-responsive subset (CD56[Bright]) (*Figure 3—figure supplement 1E–H*). Overall, these results demonstrate that circulating NK cells from CHB patients have a reduced basal mTOR activity and blunted capacity to activate this pathway upon IL-15 stimulation, a feature more evident in CD56[Bright] NK cells. This could potentially affect signalling through activating receptors. However, deregulation of mTOR did not translate into measurable basal metabolic changes with the assay we used.

## RNAseq analysis identifies an exhaustion-like signature in patient NK cells

We next took a non-hypothesis-driven approach in order to seize the molecular mechanisms responsible for NK cell deregulation in CHB patients. For this purpose, we performed RNAseq on NK cells sorted from four CHB patients and five HD using the previously described gating strategy (*Figure 1—figure supplement 2*). As shown in *Figure 4A*, principal component analysis (PCA) of the results showed a clear separation between HD and CHB patients both on PC1 and 2, with 38% and 24% variance, respectively, thus motivating further analysis of the results. Consistent with the fact that patients in this cohort are co-infected with HCMV, we found characteristics of adaptive NK cell populations in the differentially expressed genes (DEG) such as decreased *FCER1G*, *ZBTB16* (*PLZF*), or cytokine receptors mRNAs and increased *GZMH*, *KLRC4*, or *CRTAM* (*Schlums et al., 2015*). In order to work with DEG that really reflected CHB impact, we filtered out genes that were significantly regulated in adaptive NK cells, as defined in a previous study (*Schlums et al., 2015*). This process identified 253 up-regulated and 163 down-regulated genes specific of HBV infection in CHB patients (Fold Change > 2 and adjusted p-value<0.05) (*Figure 4B*). We then analysed both gene lists using the online gene annotation tool Metascape (*Zhou et al., 2019*). No significant enrichment was found in the list of down-regulated genes. In contrast, analysis of the up-regulated genes retrieved Gene annotation terms that were consistent with ongoing viral infection such as 'Viral life cycle' or

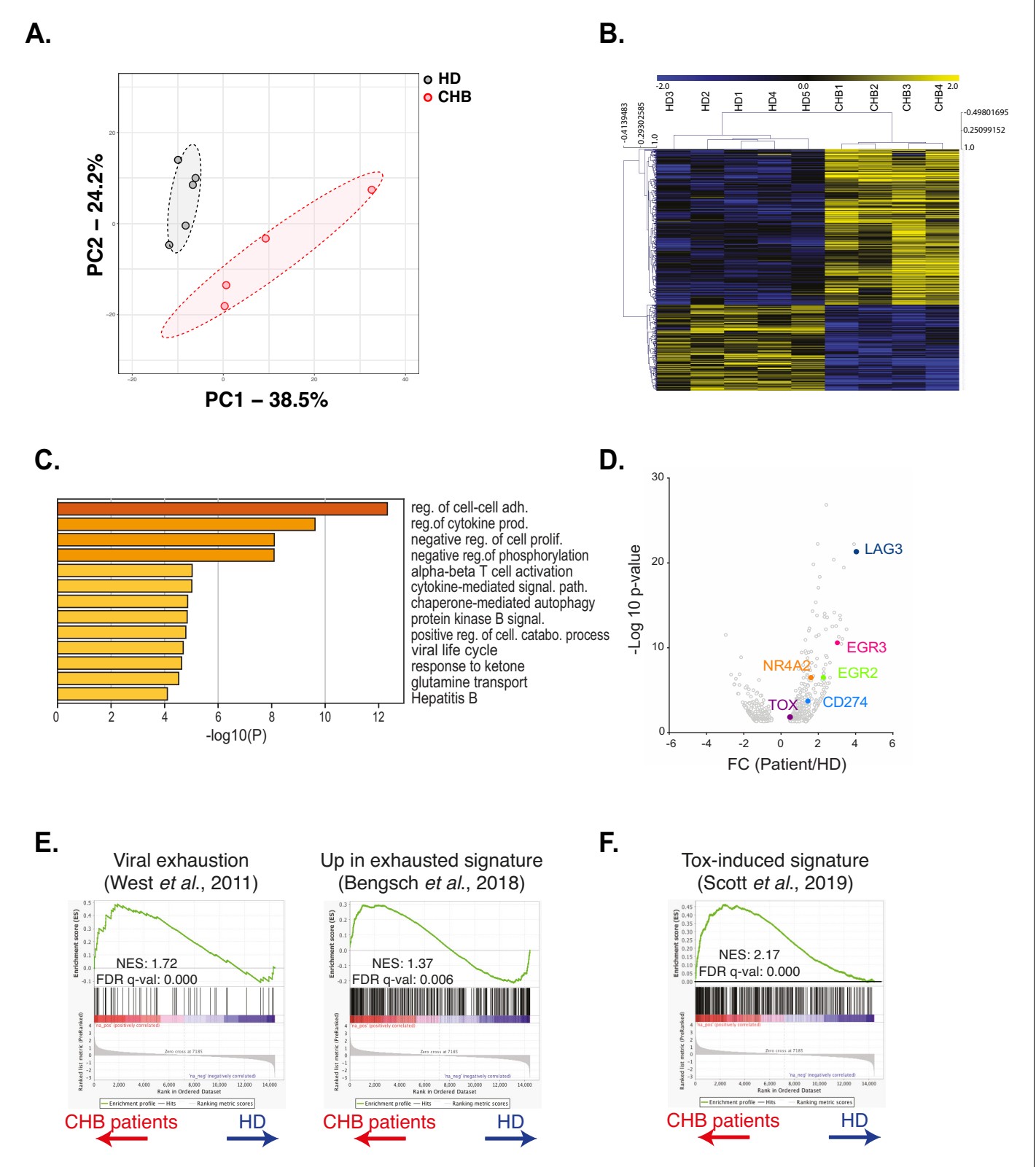

**Figure 4.** RNAseq analysis identifies an exhaustion-like signature in patient NK cells. (A) Principal component analysis of the RNAseq data is shown. (B) Heatmap of the DEG genes between HD and CHB. (C) Gene Ontology analysis of DEG up-regulated in CHB patients using Metascape. Selected terms are shown among the most significant ones. (D) Volcano plots of the DEG highlighting genes belonging to the T cell exhaustion pathway. (E,F) GSEA plots comparing HD and CHB patients are shown for the indicated gene sets. The normalised enrichment scores (NES) and FDR q-values are indicated.

*Figure 4 continued on next page*

*Figure 4 continued*

The online version of this article includes the following figure supplement(s) for figure 4:

**Figure supplement 1.** Metascape analysis of CHB patients up-regulated genes.

'Hepatitis B' (*Figure 4C*, a complete version of the analysis is given in *Figure 4—figure supplement 1*). Interestingly, some of the enriched terms referred to immune processes that are negatively impacted in NK cells of CHB patients such as 'cytokine production', 'cytokine-mediated signalling', 'phosphorylation', and 'Protein kinase B (AKT) signalling'. We also noted that 'T cell activation' was one of the enriched terms suggesting commonalities in the transcriptional regulation of NK and T cell responses. Moreover, we found that dysfunctional NK cells up-regulated several canonical genes of the T cell exhaustion program, notably immune checkpoints or their ligands, such as LAG3 and CD274 (PD-L1), or transcription factors, such as EGR2 and 3, NR4A2, and TOX (*Khan et al., 2019*; *Seo et al., 2019*; *Alfei et al., 2019*; *Scott et al., 2019*; *Yao et al., 2019*; *Barber et al., 2006*; *Williams et al., 2017*; *Chen et al., 2019*; *Figure 4D*). This observation prompted us to rigorously test whether the exhaustion transcriptional program was indeed undertaken by NK cells. To this aim, we performed gene set enrichment analysis (GSEA) using two independent datasets defined in exhausted CD8 T cells in a context of chronic viral infection (*West et al., 2011*; *Bengsch et al., 2018*). As depicted in *Figure 4E*, transcripts of these datasets were indeed strongly enriched in NK cells of CHB patients. This included TOX that we already identified among the genes significantly over-expressed in CHB patient NK cells (*Figure 4D*). This transcriptional regulator has recently been described as a key inducer of the exhausted gene signature allowing phenotypic changes and persistence of exhausted T cells (*Khan et al., 2019*; *Seo et al., 2019*; *Alfei et al., 2019*; *Scott et al., 2019*; *Yao et al., 2019*). We thus tested whether the TOX-induced gene signature was differentially expressed in HD vs CHB patients. We detected a significant enrichment of this signature in genes up-regulated in HBV patients (*Figure 4F*). In summary, NK cells of CHB patients display a transcriptional signature resembling that of exhausted T cells induced by chronic viral infections. Furthermore, our data point to the involvement of the transcription factor TOX in driving NK cell dysfunction.

## Validation of the exhausted phenotype at the protein level

In order to validate the exhausted signature at the protein level, we recruited a second cohort constituted of 12 CHB patients and 10 HD independent from the first cohort (*Table 1*). On this validation cohort, we first measured the intracellular expression of TOX in NK cells. Expression of transcription factors of the TOX family has previously been associated to early stages of NK cell development (*Yun et al., 2011*; *Aliahmad and Kaye, 2009*; *Aliahmad et al., 2010*; *Vong et al., 2014*); however, whether it is associated to the acquisition of a dysfunctional phenotype in NK cells is unknown. As shown in *Figure 5A*, NK cells from CHB patients presented higher expression of TOX validating the RNAseq data. Increased TOX expression was seen mainly in the CD56$^{dim}$ subset in CHB patients (*Figure 5—figure supplement 1*). In murine CD8 T cells, *Tox* invalidation abrogates the exhaustion program and in particular the expression of ICP such as LAG3, TIGIT, TIM3, 2B4, PD1, and CD39 (*Khan et al., 2019*; *Alfei et al., 2019*; *Scott et al., 2019*). We observed that LAG3 was the most up-regulated gene in NK cells from CHB patients (*Figure 4D*). At the protein level, we also detected a slight increase in the surface expression of LAG3 in NK cells from CHB patients compared to HD (*Figure 5B*). Of note, LAG3 and TOX protein levels were highly correlated in CHB patients specifically, further highlighting their functional link (*Figure 5C*, $R^2 = 0.80$). Despite the fact that other ICP transcripts targeted by TOX were not significantly deregulated in our RNAseq analysis, we measured their protein expression. As shown in *Figure 5D*, TIGIT was up-regulated, while TIM3 was down-regulated, and 2B4 expression was unchanged in NK cells from CHB patients compared to controls. Some CHB patients presented limited but detectable PD1 expression above the average of HD. CD39 was not expressed (data not shown). In murine CD8 T cells, the transcription factor T-BET limits the expression of PD1 (*Kao et al., 2011*). In addition, viral induced CD8 T cell exhaustion has been linked to a decrease in T-BET and an increase in the expression of the closely related T-box family transcription factor EOMES (*Paley et al., 2012*), a point also validated in human during HIV-1 infection (*Buggert et al., 2014*). It is also reported that T-BET and EOMES are

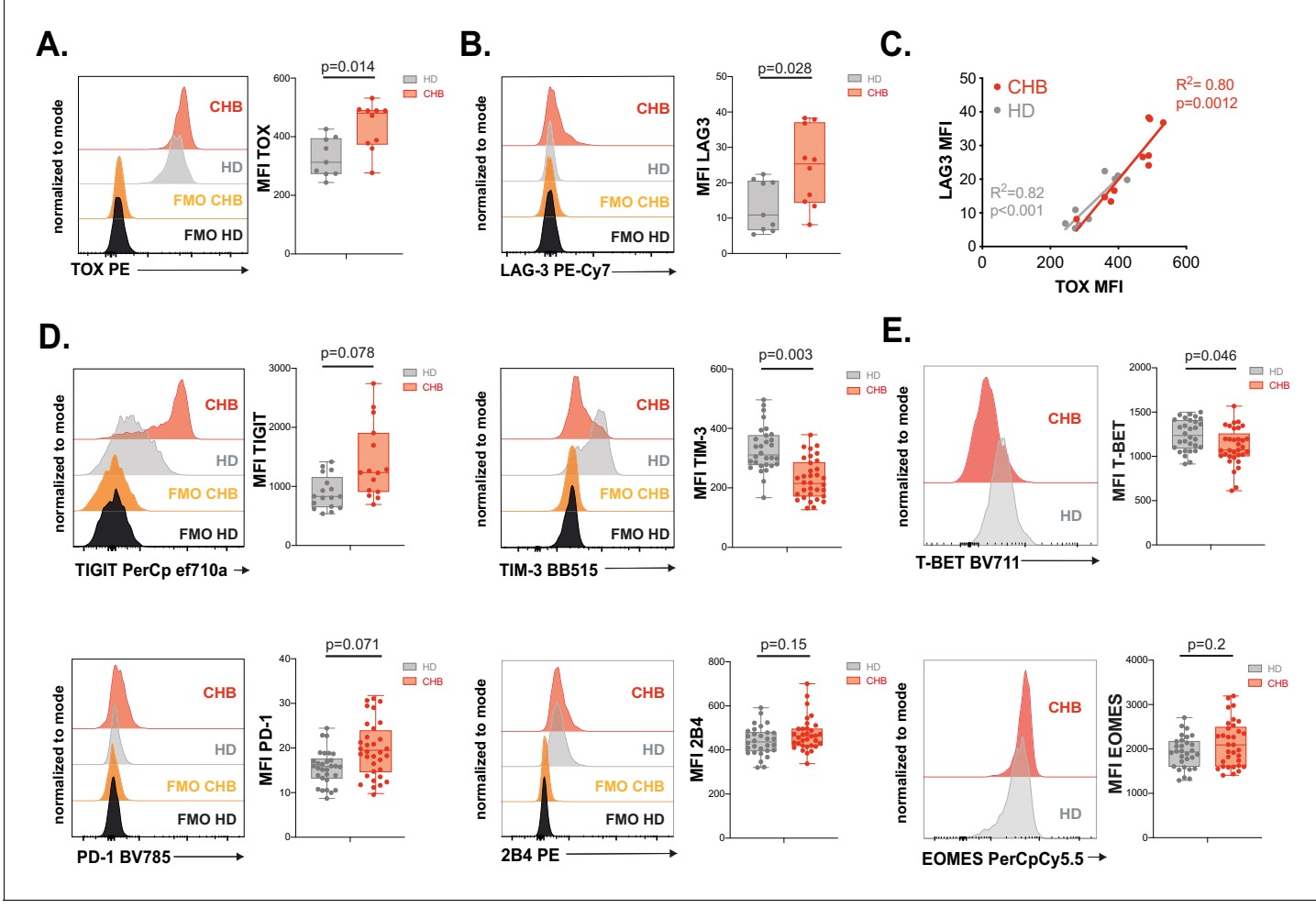

**Figure 5.** Validation of the exhausted phenotype at the protein level. (A) Intracellular staining for TOX was performed on PBMCs from 9 HD and 10 CHB samples and the MFI measured. A representative FACS histogram overlay (left panel) as well as the median MFI and individual values for each sample (right panel) are represented. (B) LAG3 expression was measured by flow cytometry on PBMCs from 9 HD and 10 CHB samples. A representative FACS histogram overlay (left panel) as well as the median MFI and individual values for each sample (right panel) are represented. (C) Linear regression plots showing the correlation between TOX MFI and LAG3 MFI using 9 HD and 10 CHB samples. The R (*Plummer et al., 2016*) and p-value calculated by linear regression are indicated. (D) ICP expression was measured by flow cytometry on NK cells from PBMCs of 17 HD and 15 CHB samples (TIGIT) or 30 HD and 32 CHB patients (TIM-3, PD-1, and 2B4). A representative FACS histogram overlay for each molecule (left panels) as well as the median MFI and individual values for each sample (right panels) are represented. (E) T-BET and EOMES expression were measured by intracellular staining on total NK cells of 30 HD and 32 CHB samples. A representative FACS histogram overlay (left panel) as well as the median MFI and individual values for each sample (right panel) are represented. Adjusted p-values are indicated on the graphs.

The online version of this article includes the following figure supplement(s) for figure 5:

**Figure supplement 1.** TOX expression in CD56[Bright] and CD56[Dim] NK cells.

both required for NK cell differentiation and acquisition of effector functions (*Zhang et al., 2018*). We observed in CHB patients that T-BET was indeed significantly down-regulated, while EOMES was unchanged (*Figure 5E*). Our data confirm the results obtained by RNAseq, demonstrating that NK cells from CHB patients display an exhaustion-associated signature at the protein level.

## RNAseq and in vitro modelling indicate that NK cell dysfunction is due to unbalanced Ca²⁺ signalling

TOX controls the expression of phenotypic characteristics in exhausted and memory T cells (*Khan et al., 2019*; *Seo et al., 2019*; *Alfei et al., 2019*; *Scott et al., 2019*; *Sekine et al., 2020*). In exhausted T cells, it is however the effector of a more global transcriptional program. The

transcription factors initiating this program appear to be NFAT-family members (*Martinez et al., 2015*; *Kallies et al., 2020*). Canonical TCR activation induces the coincident activation of $Ca^{2+}$ as well as diacylglycerol-dependent pathways, leading to activation of NFAT and AP-1 family transcription factors, respectively. It has been proposed that chronic TCR activation leads to unbalanced signal transduction predominantly activating $Ca^{2+}$-dependent signal including calcineurin activation and consequently nuclear translocation of an excess of NFAT relative to its AP-1 partners. This partnerless NFAT binds to and transactivates a specific subset of genes, distinct from the canonical subset regulated by NFAT:AP1 heterodimers and substantially overlapping the transcriptional program of exhausted cells (*Martinez et al., 2015*). In particular, transcription factors such as *TOX* or the *NR4A* family are validated targets of partnerless NFAT (*Seo et al., 2019*). Based on this knowledge, we performed GSEA using a gene set previously defined by Martinez et al., to be regulated by partnerless NFAT in context of altered signalling biased towards the $Ca^{2+}$ branch (*Martinez et al., 2015*). We found that this gene set was enriched in NK cells of CHB patients (*Figure 6A*). This suggested that, in CHB patients, NK cells are subjected to an unbalanced $Ca^{2+}$ signalling. To functionally test whether a signalling bias towards $Ca^{2+}$ would endow control NK cells with characteristics of CHB patient NK cells, we investigated the impact of ionomycin treatment on PBMCs from HD at different time points. As expected, ionomycin, a $Ca^{2+}$ ionophore, induced NFAT1 nuclear translocation as soon as 1 hr after stimulation (*Figure 6B*). This was followed by an increase in TOX expression at 4 hr (*Figure 6C*). Moreover, LAG3 was induced by ionomycin, in a way correlated to the expression of TOX (*Figure 6D*), thus recapitulating phenotypic features of CHB patients' NK cells. In order to measure the functional impact of $Ca^{2+}$ signalling on NK cells, we pre-treated PBMCs from HD for 16 hr with different concentrations of ionomycin. We then measured NK cell effector functions: degranulation and cytokine expression in response to a K562 challenge. As previously published on NK cell lines (*Romera-Cárdenas et al., 2016*), ionomycin treatment impaired NK cell effector functions (*Figure 6E*). Interestingly, at a dose of ionomycin of 100 nM, IFN-γ production was completely inhibited, while degranulation was still partially conserved, a behaviour reminiscent of the functional dichotomy observed in CHB patients. The production of MIP1-β and TNF-α showed intermediate behaviours in accordance with CHB patients' phenotype. This was achieved without negative impact on viability (data not shown). Overall, these results show that stimulation of $Ca^{2+}$ flux in isolation can reproduce key phenotypic and functional features of NK cells observed in a context of CHB infection. These data, in association with transcriptional enrichment of partnerless NFAT-dependent transcripts, strongly suggest that NK cell dysfunction observed in CHB patients is the result of unbalanced $Ca^{2+}$ signalling. This further indicates that common molecular mechanisms govern T cell exhaustion and NK cell dysfunction in contexts of chronic stimulation.

## Discussion

In the present study, we explored the impact of CHB on peripheral NK cell function and phenotype and uncovered a convergence in the transcriptional mechanisms governing T cell exhaustion and NK cell dysfunction.

We first confirmed previous reports showing that NK cell production of cytokines, chief among them IFN-γ, is blunted in response to target cell stimulation, while cytotoxic functions are not affected in the same settings (*Oliviero et al., 2009*; *Peppa et al., 2010*; *Sun et al., 2012*; *Tjwa et al., 2011*). This functional dichotomy could be particularly relevant since an early study of acute HBV infection in chimpanzee suggested that viral clearance was mediated by non-cytopathic antiviral effects (*Guidotti et al., 1999*). We can thus hypothesise that loss of cytokine-mediated control contributes to HBV escape and establishment of chronicity.

NK cells of CHB patients constituting our cohort respond normally to IL-12/18 challenge in accordance with previous findings (*Sun et al., 2012*). This is however in contrast with other reports (*Peppa et al., 2010*; *Tjwa et al., 2011*). This discrepancy could stem from differences in the abundance of HCMV-induced adaptive NK cells in the respective cohorts. Indeed, these cells present impaired responses to IL-12/18 (*Schlums et al., 2015*) and, CHB patients being frequently co-infected with HCMV (*Bayram et al., 2009*), NKG2C[+]-adaptive NK cells are significantly more represented in CHB patients (*Béziat et al., 2012*; *Schuch et al., 2019*). In our cohort, we observed that patients presenting the highest NKG2C[+] frequency were less responsive to IL-12/18 (data not

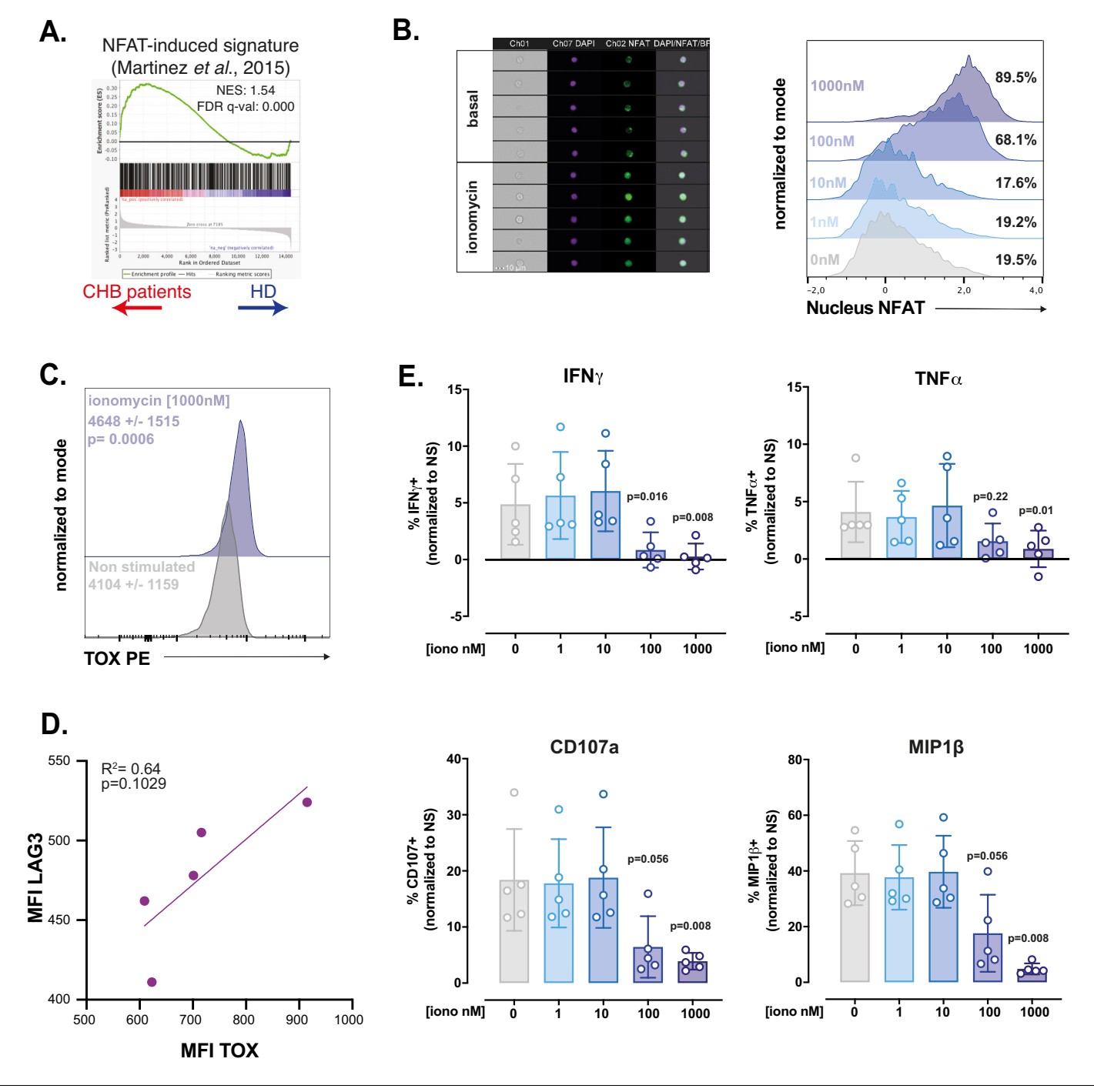

**Figure 6.** RNAseq and in vitro modelling suggest that NK cell dysfunction is due to unbalanced $Ca^{2+}$ signalling. (**A**) A GSEA plot comparing HD and CHB patients is shown for the gene set regulated by partnerless NFAT. The normalised enrichment score (NES) and FDR q-values are indicated. (**B**) PBMCs from HD were exposed to ionomycin at the indicated concentration for 1 hr. NFAT1 localisation was then analysed by image cytometry. Representative images of the transmission, nuclear (DAPI), and NFAT1 staining in NK cells at basal state or after 1 hr ionomycin treatment are shown (left panel). Histogram overlays of the parameter quantifying NFAT1 translocation are shown on the right. The experiment was performed twice. (**C**) PBMCs from HD were exposed to ionomycin for 4 hr and stained for TOX expression. A representative histogram overlay as well as the average expression ± SD is shown (n = 10). (**D**) Linear regression plots showing the correlation between TOX MFI and LAG3 MFI after O/N ionomycin treatment (1 μM) is shown. The R (*Plummer et al., 2016*) and p-value calculated by linear regression are indicated. (**E**) PBMCs from five HD were incubated with ionomycin at the indicated concentration for 16 hr. NK cell capacity to degranulate or produce the indicated cytokines upon K562 stimulation was then *Figure 6 continued on next page*

*Figure 6 continued*

measured by flow cytometry. The average (± SD) and individual values of the proportion of positive NK cells normalised to the non-stimulated condition is shown. The experiment has been performed twice. Exact p-values are indicated on the graphs.

shown). Interestingly, the overall functionality of HCMV-induced adaptive NK cell is not affected by HBV (*Schuch et al., 2019*). It thus seems that both viruses engage non-overlapping NK cell subsets.

We report here that the functional impairment imposed by CHB significantly imprints the NK cell transcriptome. A recent study was performed on HBV-infected patients from all four phases of infection (*Boeijen et al., 2019*). Very few transcripts were significantly deregulated in the inactive carriers compared to HD controls. In particular, no IFN-I-stimulated genes were up-regulated, a finding we confirm with our dataset. Further mining our data, we found that the transcriptome of NK cells in CHB patients contained gene signatures defined in exhausted CD8 T cells arising in chronic viral infection models. This strongly indicates that a mechanism akin to exhaustion is at play to explain the dysfunctional phenotype of NK cells in CHB patients. In particular, we found that the gene signatures defined by transcripts regulated by the transcription factor TOX were enriched in CHB patients. This transcription factor has recently been involved in the establishment of the exhausted phenotype in CD8 T cells, its expression being driven by chronic TCR stimulation (*Khan et al., 2019*; *Seo et al., 2019*; *Alfei et al., 2019*; *Scott et al., 2019*; *Yao et al., 2019*). Of note, its association with functional impairment has recently been called into question. Indeed, TOX is highly expressed in recently or chronically stimulated but fully functional human CD8 T cells in a series of viral infections (*Sekine et al., 2020*). In this setting, TOX is still associated with the expression of phenotypic markers such as ICPs. Similarly, in our cohort, TOX expression positively correlated with LAG3 expression, but not with the decrease in functional capacity (data not shown). It thus seems that TOX plays a role in instructing the phenotype of NK cells rather than their functional capacities in CHB patient. These results further underscore the ontogenic and functional proximity of NK cells and CD8 T cells. Monitoring reversal of NK cell dysfunction in therapeutic settings aimed at reversing T cell exhaustion in chronic diseases should thus be considered. Mechanistically, it further suggests that direct chronic stimulation, perhaps through NK activating receptors, is responsible for the dysfunction. Reinforcing this hypothesis is the fact that we and others observed decreased NKp30 and CD16 expression on CHB patient NK cells suggesting chronic engagement (*Peppa et al., 2010*; *Tjwa et al., 2011*). The nature of the exact stimulus and receptor involved remains to be deciphered. Since we studied peripheral blood NK cells, a population largely distinct from liver resident NK cells (*Marotel et al., 2016*), it is unlikely that dysfunction results from direct stimulation by infected cells. Circulating HB antigen is however present in complex with neutralising antibodies (*Madalinski et al., 1991*). These immune complexes could chronically stimulate NK cells through CD16 cross-linking. In this respect, it has recently been reported that influenza vaccination increases NK cell function inducing memory-like differentiation (*Goodier et al., 2016a*). This property is based on the sensing of immune complexes by CD16 (*Goodier et al., 2016b*). Based on the CD8 T cell system where a given stimulus can give rise to functional memory cells in acute settings and to exhausted cells if it becomes chronic (*Wherry and Ahmed, 2004*; *Wherry et al., 2007*), we can hypothesise that chronic stimulation of NK cells through CD16 progressively drives the exhausted signature seen in our RNAseq.

Given the molecular proximity we describe, we could expect exhausted NK cells to display other characteristics of exhaustion observed in T cells. A key feature when T cells progress to exhaustion is a defective metabolism and blunted activation of the mTOR pathway, a major coordinator of anabolic and catabolic pathways. In CHB patients at basal state and in response to IL-15, we indeed observed a decrease in the activation of the mTOR pathway, quantified by lower mTORC1 and 2 activities. Importantly, this defect selectively affected the mTOR pathway since phosphorylation of STAT5, another signalling event downstream IL-15 Receptor (IL-15R), was normal. Such a depression in mTOR activity due to chronic activation is reminiscent of the situation that prevails in so called 'non-educated' or 'disarmed' hyporesponsive NK cells (*Marçais et al., 2017*). Of note, this similarity also points towards excessive stimulation of activating receptors, the mechanism envisaged to explain NK cell disarming (*Goodridge et al., 2019*; *Wu and Raulet, 1997*; *Johansson et al., 1997*). How chronic stimulation leads to decreased mTOR activity remains to be investigated. Weaker mTORC1 activity could lead to impaired cell metabolism. However, we did not detect any significant

metabolic defect in CHB patients using several readouts, in contrast to what has been reported for CD8 T cells in CHB patients (*Fisicaro et al., 2017*). This finding, associated with the fact that the cytotoxic capacity is spared and that expression of ICPs is low in most patients, suggests either that exhaustion is at an early stage or that, unlike T cells, NK cells have a limited capacity to engage a full exhaustion program, perhaps as a result of a more limited half-life.

At the molecular level, our results point to a major role of $Ca^{2+}$ and downstream NFAT signalling in the induction of NK cell dysfunction. Indeed, we detected a transcriptional signature indicative of improper activation of NFAT transcription factors in dysfunctional NK cells from CHB patients. In line with our findings, we devised an in vitro model based on unbalanced NK cell activation by activation of the $Ca^{2+}$ pathway alone using ionomycin. We found that ionomycin increased TOX expression and induced LAG3. Keeping in mind that dysfunction is only partial in CHB patients, we titrated ionomycin and found that a dose of ionomycin inducing suboptimal NFAT1 activation in NK cells resulted in a complete loss of their capacity to produce IFN-γ, while degranulation was partially conserved. This in vitro model thus presents several points of convergence with the phenotypic and functional characteristics of NK cells in our CHB cohort. Of note, ionomycin treatment also induced hypo-responsiveness in T cells and NK cell lines reinforcing the parallel between both cell types (*Romera-Cárdenas et al., 2016*; *Macián et al., 2002*; *Dubois et al., 1998*). Interestingly, the effect of ionomycin treatment can be adjusted, so that higher concentration leads to more pronounced defects.

How an imbalance in $Ca^{2+}$ responses is triggered in CHB context remains an open question. In the T cell field, such an imbalance is proposed to be the result of defective co-stimulation (*Martinez et al., 2015*). However, in NK cells, the conceptual framework of stimulatory vs costimulatory receptors is not as clearly established. We hypothesise that a mechanism similar to a recently proposed model of disarming could be at play (*Goodridge et al., 2019*). In this model, continuous leakage of cytotoxic granules in response to chronic activating receptor triggering is involved. Since cytotoxic granules are part of the acidic $Ca^{2+}$ stores, we can hypothesise that low-grade degranulation would lead to increased concentration of intracellular free $Ca^{2+}$ and sequential activation of calcineurin and NFAT. However, in contrast to CHB, disarming does not imprint the transcriptome of circulating NK cells (*Goodridge et al., 2019*; *Guia et al., 2011*). It is thus likely that a combination of signals such as stimulation of activating receptors and increased levels of anti-inflammatory cytokines (TGF-β and IL-10) (this study and others *Peppa et al., 2010*; *Sun et al., 2012*) shapes the NK cell phenotype in CHB patients.

In addition to their interest at the basic level, the findings we present here could aid to rationally design successful NK cell reinvigoration strategies that could contribute to viral elimination. Based on our findings, we would hypothesize that targeting the $Ca^{2+}$ pathway or mechanistically linked molecules to re-establish a correct signalling balance could have a positive impact on effector functions. In this respect, a recent screen identified ingenol mebutate, an activator of PKCs, for its ability to revert T cell exhaustion (*Marro et al., 2019*). Mechanistically, this compound was able to complement ionomycin signal for efficient reinvigoration of virus-specific T cells. Indeed, PKCs are activated by a signalling branch parallel to $Ca^{2+}$ signalling and participate in the activation of the transcription factor AP1, the missing partner of NFAT in exhausted cells (*Baier-Bitterlich et al., 1996*). PKC activation could thus compensate the observed signalling defect and constitute a useful target to restore dysfunctional NK cells activity. More studies will be required to address this point.

In summary, we provide evidence that dysfunctional NK cells of CHB patients present a molecular signature similar to the one of exhausted T cells at the transcriptional and protein level. This signature is indicative of a signalling imbalance involving calcium. This could open the way towards original therapeutic targets.

## Materials and methods

### Patients and healthy donors

Peripheral blood samples from healthy subjects were obtained from the French blood agency (Etablissement Français du sang, Lyon and Limoges, France). PBMCs from CHB patients and clinical assessments were obtained during routine hepatitis consultations. All participants provided written informed consent in accordance with the procedure approved by the local ethics committee (Comité

de Protection des Personnes, Centre Hospitalier Universitaire de Limoges, Limoges, France) and the Interventional research protocol involving human samples (Code promotor LiNKeB project: 87RI18-0021). Patients and healthy donor characteristics are detailed in *Table 1*. All patients were diagnosed as inactive carriers according to the American Association for the Study of Liver Diseases (AASLD) guidelines for treatment of CHB (*European Association for the Study of the Liver. Electronic address: easloffice@easloffice.eu and European Association for the Study of the Liver, 2017*). HCMV seropositivity status of patients and healthy donors was determined by ELISA (Hôpital de la Croix-Rousse, Lyon, France).

## PBMC isolation

Human PBMCs were separated from peripheral blood by Ficoll gradient centrifugation (Eurobio Laboratoires et AbCys) at room temperature (RT). Cells were then resuspended in heat-inactivated FCS with 20% DMSO, progressively cooled down to −80°C, and stored in cryotubes in liquid nitrogen.

## Cell culture and treatments

Cells (ATCC) were cultured in RPMI 1640 medium (Invitrogen Life Technologies) supplemented with 10% of FCS, 2 mM L-glutamine, 10 mM of penicillin/ streptomycin (HCL Technologies), 1 mM sodium pyruvate (PAA Laboratories), and 20 mM HEPES (Gibco). Cells were regularly tested for mycoplasma and were negative. For phosphorylated protein analysis, cells were stimulated with IL-15 (Peprotech, 100 ng/mL) during 30 min at 37°C. In some experiments, PBMCs were treated with increasing doses of ionomycin (Sigma) during 16 hr.

## Flow cytometry analysis

PBMCs were rapidly thawed in medium heated to 37°C and kept overnight at 4°C. Cells were then immunostained during 30 min at 4°C with the appropriate monoclonal antibodies detailed in *Supplementary file 1*. Intracellular staining of transcription factors and cytotoxic molecules was performed with Foxp3 Fixation/Permeabilisation concentrate and diluent (eBioscience). Intracellular staining of cytokines and chemokines was performed with Cytofix/Cytoperm (BD Biosciences). Intracellular staining of phosphorylated proteins was performed with Lyse/Fix and Perm III buffers (BD Biosciences). Phosphorylated proteins were then stained during 40 min at RT. Kynurenine uptake was measured as previously described (*Sinclair et al., 2018*). Briefly, PBMCs were stained for surface markers for NK cell identification and resuspended in phosphate-buffered saline (PBS). Baseline fluorescence in the BV421 channel was recorded for 30 s, and kynurenine (200 μM final concentration) was added before a further 4 min acquisition. The assay was run at 37°C. Flow cytometric analysis was performed on LSR Fortessa 5L (Becton-Dickinson). Fluorescence Minus One controls were used to set the gates, and data were analysed with FlowJo 10.5.0 software (Tree Star). Gating strategy is presented is *Figure 1—figure supplement 2*.

## Image stream analysis

PBMCs from HD were stimulated for 1 hr with ionomycin at the indicated concentration. They were then stained with a fixable viability dye (ThermoFisher Scientific) and stained for surface markers allowing NK cell identification (CD3, CD19, CD14, and CD56). The samples were then fixed and permeabilised using the Foxp3 Fixation/Permeabilisation concentrate and diluent (eBioscience) and stained with an anti-NFAT1 (Cell Signaling Technology). Sample acquisition was made on an Image-Stream X Mark II (Amnis-EMD Millipore, Darmstadt, Germany) with ×40 magnification and analyzed with IDEAS software (v6.0).

## NK cell stimulation

Human PBMCs were stimulated with recombinant human IL-12 (Peprotech) and IL-18 (R and D Systems) at 10 ng/mL each or co-cultured for 4 hr with K562 cells or with Granta cells coated with Rituximab at a 1.1 ratio in the presence of Golgi Stop (BD Biosciences). The percentage of NK cells positive for CD107a, MIP1-β, IFN-γ, and TNF-α was then determined by flow cytometry.

## Cytotoxic assay

Human PBMCs were rapidly thawed in medium heated to 37°C and kept overnight at 4°C. Cells were then co-cultured for 4 hr at different Effectors:Targets ratios with K562 NanoLuc[+32]. Supernatant was collected, and bioluminescence was measured using a TECAN Instrument luminometer after addition of Furimazin, the NanoLuc substrate (Promega). Furimazin was generated from Hikarazin as previously described (*Coutant et al., 2019*; *Coutant et al., 2020*).

## Mitochondria analysis

Human PBMCs were incubated with MitoSOX Red (5 µM) and Mitotracker Green (1 µM) (both from Molecular Probes, Life Technologies) in PBS during 10 min at 37°C before flow cytometry extra-cellular staining.

## Serum TGF-β1 quantification

Active TGF-β1 serum levels in patients and healthy donors were measured using LEGEND MAX Free Active TGF-β1 ELISA Kit with pre-coated plates (Biolegend). The assay was run according to the manufacturer's recommendations.

## RNA sequencing

NK cells from five HD and four CHB patients were sorted as live/dead$^-$/CD4$^-$/14$^-$/19$^-$/CD3$^-$/CD56$^+$ cells by flow cytometry. Samples from HD were sorted in a BSL2 cytometry platform (Anira cytométrie, SFR Biosciences, Lyon, France), whereas samples from CHB were sorted in a BSL3 cytometry platform (Toulouse, France). NK cells were then lysed in Direct-Zol (Ozyme). The RNA libraries were prepared according to the protocol of *Picelli et al., 2014*. Total RNA was purified using the Direct-Zol RNA Microprep Kit (Ozyme) according to the recommendations provided and was quantified using the QuantiFluor RNA system (Promega). One microlitre of 10 µM of oligo-dT primer and 1 µl of 10 µM of dNTPs were added to 0.3 ng of total RNA in a final volume of 2.3 µl. The Oligo-dTs were hybridized for 3 min at 72°C, and a reverse transcription reaction was carried out as described in the Nature protocols. The complementary DNAs (cDNAs) were purified on AmpureXP beads (Beckman Coulter), and the quality was checked on a D5000 screening strip and analysed on a 4200 strip station (Agilent). Three nanograms of cDNA was labelled using the Nextera XT DNA sample preparation kit (Illumina). The labelled fragments were then amplified following PCR cycles and purified on AmpureXP beads (Beckman Coulter). The quality of the bank was checked on a D1000 screen tape and analysed on a 4200 tape station (Agilent). The sequencing of the banks was carried out by the GenomEast platform, member of the 'France Génomique' consortium (ANR-10-INBS-0009).

## In silico analyses

PCA was performed using the R software (version 3.6.1) after data normalisation and graphed with the ggplot2 package. We then obtained a first DEG list using an adjusted p-value<0.05 as a cutoff (750 genes). In order to keep DEG deregulated only by HBV infection in our subsequent analysis, we removed the ones known to be affected by HCMV infection. To identify genes expressed by NK cells and affected by HCMV infection, we used a previously published microarray dataset comparing conventional and adaptive NK cells (*Schlums et al., 2015*). In this HCMV dataset, DEG were identified as genes presenting a p-value below 0.005 (1357 genes in total). Elements common to the two DEG lists and that did not satisfy the relation *abs(logFC(HBV))>2\*abs(logFC(HCMV))* were subtracted from the list of DEG obtained in our RNAseq study. We further identified genes showing a *FC(HBV) >2* to obtain the final list of DEG genes. This DEG list was used for heatmap and Metascape analysis. The heatmap was constructed using Multiple Experiment Viewer with Row Median centring of the data (*Saeed et al., 2003*). Functional annotations of DEG were performed with Metascape (*Zhou et al., 2019*) using default parameters. Regarding GSEA, indicated gene sets of publicly available expression data were obtained. To statistically test whether these gene sets were enriched in specific conditions, we performed pairwise comparisons between HD and CHB patients' conditions using the GSEA method (http://www.broad.mit.edu/gsea).

## Statistical analysis

Clinical data were processed with the R statistical environment. After cleaning empirical data from outliers, we transformed to log scale the parameters that were showing log-normal distribution and kept others unchanged. Then we used a generalised linear model with binomial family (logistic regression model) in simple way, that is one regressor at time, to quantify for each biological parameter independently the probability to be linked to our outcome variable ( = to be HD or CHB). The results for all parameters were taken together to correct for the multiple testing using the Benjamini–Hochberg method. Graphical representations were done using Prism 5 (Graph-Pad Software) unless otherwise stated.

## Acknowledgements

We thank the SFR Biosciences (UMS3444/CNRS, ENSL, UCBL, US8/INSERM) facilities, in particular the Plateau de Biologie Expérimentale de la Souris, and the flow cytometry facility. We would like to thank Dr PO Vidalain who provided the K562 NanoLuc, Dr Y Janin who provided the NanoLuc substrate, and Dr. Christophe Ramière for the determination of HCMV status in Hôpital Croix-Rousse. We also acknowledge the contribution of the GenomEast platform from Strasbourg, France. TW lab is supported by the Agence Nationale de la Recherche (ANR JC *SPHINKS* to TW and ANR JC *BaNK* to AM), the ARC foundation (équipe labellisée), the European Research council (ERC-Stg 281025), and receives institutional grants from the Institut National de la Santé et de la Recherche Médicale (INSERM), Centre National de la Recherche Scientifique (CNRS), Université Claude Bernard Lyon1 and ENS de Lyon. MM is the recipient of a fellowship from La Ligue Nationale contre le Cancer. This work was supported by l'Agence Nationale de Recherche sur les SIDA et les Hépatites virales (ANRS projects, ECTZ22398, ECTZ11169 and ECTZ19856 to UH) and Comité du Rhône LNCC (to UH).

## Additional information

### Funding

| Funder | Grant reference number | Author |
|---|---|---|
| Agence Nationale de Recherches sur le Sida et les Hépatites Virales | ECTZ22398 | Uzma Hasan |
| Agence Nationale de Recherches sur le Sida et les Hépatites Virales | ECTZ11169 | Uzma Hasan |
| Agence Nationale de Recherches sur le Sida et les Hépatites Virales | ECTZ19856 | Uzma Hasan |
| Agence Nationale de la Recherche | BaNK | Antoine Marçais |
| Agence Nationale de la Recherche | SPHINKS | Thierry Walzer |
| Association de Recherche sur le Cancer | Equipe labellisée | Thierry Walzer |
| H2020 European Research Council | ERC-Stg 281025 | Thierry Walzer |
| Ligue Contre le Cancer | Graduate student fellowship | Marie Marotel |
| La Ligue du Rhone | LNCC | Uzma Hasan |

The funders had no role in study design, data collection and interpretation, or the decision to submit the work for publication.

### Author contributions

Marie Marotel, Conceptualization, Formal analysis, Investigation, Methodology, Writing - original draft, Writing - review and editing; Marine Villard, Annabelle Drouillard, Formal analysis,

Investigation, Writing - review and editing; Issam Tout, Marine Pujol, Michelle Ainouze, Guillaume Roblot, Melissa Gomez, Investigation; Laurie Besson, Yamila Rocca, Investigation, Writing - review and editing; Omran Allatif, Formal analysis; Sébastien Viel, Supervision, Writing - review and editing; Veronique Loustaud, Resources, Writing - review and editing; Sophie Alain, David Durantel, Resources; Thierry Walzer, Conceptualization, Supervision, Funding acquisition, Project administration, Writing - review and editing; Uzma Hasan, Conceptualization, Funding acquisition, Project administration, Writing - review and editing; Antoine Marçais, Conceptualization, Supervision, Funding acquisition, Writing - original draft, Project administration, Writing - review and editing

### Author ORCIDs
Marie Marotel ⓘD https://orcid.org/0000-0002-1618-7602
Sébastien Viel ⓘD https://orcid.org/0000-0002-5085-443X
Thierry Walzer ⓘD https://orcid.org/0000-0002-0857-8179
Antoine Marçais ⓘD https://orcid.org/0000-0002-3591-6268

### Ethics
Human subjects: All participants provided written informed consent in accordance with the procedure approved by the local ethics committee (Comité de Protection des Personnes, Centre Hospitalier Universitaire de Limoges, Limoges, France) and the Interventional research protocol involving human samples (Code promotor LiNKeB project: 87RI18-0021).

### Decision letter and Author response
Decision letter https://doi.org/10.7554/eLife.60095.sa1
Author response https://doi.org/10.7554/eLife.60095.sa2

## Additional files

### Supplementary files
• Supplementary file 1. Antibodies used.

• Transparent reporting form

### Data availability
Sequencing data have been deposited in GEO under accession codes GSE153946.

The following dataset was generated:

| Author(s) | Year | Dataset title | Dataset URL | Database and Identifier |
|---|---|---|---|---|
| Marotel M | 2023 | RNAseq | https://www.ncbi.nlm.nih.gov/geo/query/acc.cgi?acc=GSE153946 | NCBI Gene Expression Omnibus, GSE153946 |

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
