## [Decision Letter]

**Acceptance summary:**

It is well established that NK cells have impaired functions in chronic hepatitis B (CHB) patients, but the molecular mechanism behind this dysfunction remains poorly characterized. The present study demonstrated that peripheral NK cells in CHB patients display several characteristics of exhausted T cells and defective mTOR pathway activity. Analysis of NK cell transcriptome in CHB patients revealed enrichment for transcripts expressed in exhausted T cells suggesting that NK cell dysfunction and T cell exhaustion share some common mechanisms. In particular, the transcription factor TOX and its targets were over-expressed in NK cells of CHB patients. Stimulation of the calcium-dependent pathway recapitulated features of NK cells from CHB patients, indicating that deregulated calcium signaling could be a central event in NK cell dysfunction occurring during chronic infections, similar to induction of T cell exhaustion. Collectively, the results of this study demonstrate that circulating NK cells in CHB patients exhibit key molecular features reminiscent of T cell exhaustion.

**Decision letter after peer review:**

Thank you for sending your article entitled "Peripheral Natural Killer cells from chronic hepatitis B patients display molecular hallmarks of T cell exhaustion" for peer review at *eLife*. Your article is being evaluated by Satyajit Rath as the Senior Editor, a Reviewing Editor, and three reviewers.

Essential revisions:

1) Figure 3 – the authors focus on CD56bright NK cells when measuring mTOR activation, as CD56bright NK cells are more responsive to IL-15. They show that in HBV patients CD56bright NK cells have impaired response to mTOR activation. They correlate this finding with several metabolic parameters in total NK cells. Since CD56bright NK cells represent only a small fraction of NK cells it is not clear why the metabolic parameters were not analyzed only on the CD56bright population as well, or vice versa, why the total NK cells were not compared in both cases (mTOR activation and metabolic characteristics). At the current state, no conclusion can be reached by comparing these two sets of data. Also, it is not clear if cells that have reduced ability to activate mTOR upon IL-15 stimulation contribute to other observations presented, e.g. if this finding would explain reduced NK cell ability to produce IFNγ, changes in NK phenotype or transcriptome.

2) Several metabolic parameters are studied, however, it is not clear how they were selected as there are many other metabolic processes involved in NK cell response which could be important and deregulated in CHB. Besides, the only basal metabolic state was analyzed, but it remains unclear if CHB NK cells show the same metabolic characteristics upon activation.

3) Figure 5 – isotype controls are missing in all histograms. The authors state in the text “Increased TOX expression was seen mainly in the CD56dim subset in CHB patients.”, however, they do not provide data for this statement. As mentioned previously, the effects of CHB on NK mTOR signaling are the highest in the CD56bright population, so it is not clear how these data do relate one to each other.

4) The authors provide evidence that expression of transcription factor TOX is increased and T-bet expression is reduced to support the transcriptome data on the similarity of CHB NK cells and exhausted CD8^+^ T cells. However, they do not provide the evidence on the co-expression of these transcription factors, and if their changed expression directly correlates with reduced functional properties of NK cells, e.g. if NK cells having high TOX and low T-bet will produce less IFNγ.

5) To address their hypothesis on NFAT involvement in NK cell exhaustion and TOX expression the authors stimulate NK cells in vitro with ionomycin and show that pre-treatment with ionomycin renders NK cells hyporesponsive. They titrate the effect of ionomycin and find an ionomycin concentration which is inducing a reduction of IFNγ response without affecting degranulation. While the reduction of IFNγ response in this experiment is observed as in chronic HBV infection, this model should be validated before making any claims. It is not clear how treating with a higher concentration of ionomycin can mimic NK cell exhaustion that occurs over months or years. Theoretically, it cannot be a transient over-flux of calcium that initiates the expression of TOX and leading to NK cell exhaustion. NFAT/Calcineurin could play a role in the formation of NK cell exhaustion. However, the over-activation of NK cells from healthy control does not prove that this mechanism is the cause of the pathological outcome. Also, cells from patients might be investigated. Also, besides the ionomycin treatment performed, calcium flux experiment in sorted cells based on the phenotypes described would have been elegant.

6) Given significant differences between the published characteristics of T cell exhaustion and findings described in the current work, it is not appropriate to conclude that the results are similar. This applies to both phenotypic and functional changes. For example, in multiple viral infection models, the decrease in IFN-γ production occurs in a step-wise manner during the progress of T cell exhaustion. In the current work, the authors show a significant and complete reduction of IFN-γ production in all the patients analyzed. Importantly, the number of T cells that produce multiple cytokines such as IFN-γ and TNF-α are reduced. However, it does not appear that these two cytokines are concurrently reduced in *Hep*-B patients. Another difference is that the NK cells from *Hep*-B patients are able to mediate normal cytotoxicity against K562 cells while the exhausted T cells are impaired in mediating this effector function. While it may be true that the NK cells in the *Hep*-B patients undergoing exhaustion, it may not be appropriate to equate this phenomenon with that seen in T cells.

7) The link that the authors are providing between mTOR-S6-NK cell exhaustion is not clear. The reduction in the phosphorylation of AKT is significant; but, moderate. Does this physiologically relevant? Does the alternate pathway mediated by PIM kinases is the one primarily affected in the NK cells from the Heo-B patients?

8) Apart from NFAT, T-bet, BATF, EOMES, FOXO1, BLIMP1, and IRF4 have been implicated in playing a significant role in causing T cell exhaustion. What are the reasons that the gene signatures representing these transcription factors did not come through from the RNA sequencing analyses? In addition, a large part of the manuscript relates to TOX and its involvement in exhaustion. However, a recent article (Sekine et al., 2020) demonstrated that TOX is expressed by most circulating effector memory CD8^+^ T cell subsets and not exclusively linked to exhaustion. This is an important piece of work where such data might be integrated and invite to reinterpret results and conclusions.

9) Figure 2 – as it seems, results display total NK cells which makes sometimes differences difficult to interpret, if possible, please provide in the supplement at least phenotype of Bright vs DIM NKG2A+ vs DIM NKG2A-.

10) Phosflow as well as mitochondrial analysis are always difficult to perform due to technical specificities, efficient detection of epitopes, atypical fluorescence leakages, or analysis of small shift differences. For both techniques, in order to highlight the quality of the datasets, please provide representative histograms as well as positive and negative controls, and gating strategy to further convince the readers.

---

## [Author Response]

Essential revisions:1) Figure 3 – the authors focus on CD56bright NK cells when measuring mTOR activation, as CD56bright NK cells are more responsive to IL-15. They show that in HBV patients CD56bright NK cells have impaired response to mTOR activation. They correlate this finding with several metabolic parameters in total NK cells. Since CD56bright NK cells represent only a small fraction of NK cells it is not clear why the metabolic parameters were not analyzed only on the CD56bright population as well, or vice versa, why the total NK cells were not compared in both cases (mTOR activation and metabolic characteristics). At the current state, no conclusion can be reached by comparing these two sets of data.

We thank the reviewer for this comment and agree with the fact that comparing different NK cell populations is misleading. To address this issue, we now provide a detailed analysis of all phospho-epitopes and metabolic parameters in total NK cells (Figure 3) and in CD56^Dim^ or CD56^Bright^ subsets (Figure 3—figure supplement 1).

Also, it is not clear if cells that have reduced ability to activate mTOR upon IL-15 stimulation contribute to other observations presented, e.g. if this finding would explain reduced NK cell ability to produce IFNγ, changes in NK phenotype or transcriptome.

We thank the reviewer for allowing us to clarify this point. We previously showed in murine models that mTOR activity and NK cell functionality were tightly correlated, mTOR activity behaving as a molecular rheostat of ITAM signaling (Marçais et al., 2017). In the present study, there is a correlation between pS6 level at basal state and after IL-15 stimulation and the proportion of IFN-γ^+^ NK cells following K562 stimulation on a patient-per-patient basis in our cohort (See Figure 3—figure supplement 1D in the revised version of the manuscript and Author response image 1). We now show this piece of data in the revised version of the manuscript (Figure 3—figure supplement 1D).

**Author response image 1. respfig1:** Correlation between percentages of IFN-γ+ following K562 stimulation and pS6 level following IL-15 stimulation. PBMCs from HD (n=30) or CHB patients (n=32) were co-cultured with K562 during 4 hours and the proportion of NK cells expressing IFN-γ was determined by immunostaining. The same samples were stimulated with IL-15 100ng/ml for 30 minutes and pS6 level was measured after phospho-epitope staining. The result of both readouts were correlated on a patient by patient basis.

Regarding phenotypic parameters, pS6 level was positively correlated to CD160 (adjusted pvalue=0.02, correlation coefficient 0.37). None of the other correlations were significant.

2) Several metabolic parameters are studied, however, it is not clear how they were selected as there are many other metabolic processes involved in NK cell response which could be important and deregulated in CHB. Besides, the only basal metabolic state was analyzed, but it remains unclear if CHB NK cells show the same metabolic characteristics upon activation.

Due to the limited number of NK cells that can be isolated from patient samples and the fact that analysis of live cells from CHB patients requires A3 safety level according to French regulation, we were limited to assays that can be analyzed by flow-cytometry. In the first version of the manuscript, we limited ourselves to the most common ones (morphology, transporters levels) and some that have been consistently shown to be affected in processes related to exhaustion (mitochondria parameters). We performed these assays at basal state to stay as close as possible to the physiological state. We now repeated these assays following overnight IL-15 stimulation, to shed light on the metabolic behavior of CHB NK cells upon activation. In addition, we now added kynurenine uptake as a more direct way to assess AA uptake capacity (Sinclair et al., 2018). The new data are now presented in the Results section (Figure 3C-G and Figure 3—figure supplement 1E-H).

3) Figure 5 – isotype controls are missing in all histograms.

According to a recently published set of guidelines for flow-cytometry (Cossarizza et al., 2019), isotype controls are of limited use due to unspecific binding and characteristics such as difference in antibody concentration or ratio of coupling to the fluorochromes. Therefore, we routinely perform Fluorescence Minus One (FMO). These have now been added in the different histograms shown in Figure 5. For the reviewer, we also performed isotype controls, the results are depicted below (Author response image 2) and can replace the current version of the data if the reviewers deem appropriate.

**Author response image 2. respfig2:** Representative histogram plots showing the expression level of the indicated markers as well as FMO and isotype control stainings on NK cells of HD or CHB patients.

The authors state in the text “Increased TOX expression was seen mainly in the CD56dim subset in CHB patients.”, however, they do not provide data for this statement.

These data are now shown in the revised version of the manuscript (Figure 5—figure supplement 1).

As mentioned previously, the effects of CHB on NK mTOR signaling are the highest in the CD56bright population, so it is not clear how these data do relate one to each other.

We thank the reviewer for the opportunity to clarify the following point: the effect of CHB on mTOR signaling are easier to detect on the CD56^Bright^ population upon IL-15 stimulation as this is the most sensitive one to IL-15. Therefore, the amplitude of the phosphostaining between non- stimulated and stimulated is the highest in CD56Bright, thus maximizing the chances of detecting a difference (See Author response image 3). However, at basal state, mTOR signaling is also affected in the total NK cell population (Figure 3A) and in the CD56^Dim^ population (Figure 3—figure supplement 1C) in CHB patients. This has been clarified in the revised version of the manuscript.

**Author response image 3. respfig3:** Correlation between pS6 level in CD56Dim and CD56Bright NK cells following IL-15 stimulation. PBMCs from HD (n=30) or CHB patients (n=32) were stimulated with IL-15 at 100 ng/mL for 30 min, prior to phospho-epitope staining (pS6 Ser235/236). MFI values for each individual are represented.

4) The authors provide evidence that expression of transcription factor TOX is increased and T-bet expression is reduced to support the transcriptome data on the similarity of CHB NK cells and exhausted CD8^+^ T cells. However, they do not provide the evidence on the co-expression of these transcription factors, and if their changed expression directly correlates with reduced functional properties of NK cells, e.g. if NK cells having high TOX and low T-bet will produce less IFNγ.

These questions are indeed very interesting, and we thank the reviewer for giving us the opportunity to investigate them. We costained for T-BET and TOX that are indeed co-expressed and expressed in a unimodal manner (See Author response image 4). We could not detect functional differences upon gating on TOX^high^/T-BET^low^ vs TOX^low^/T-BET^high^ cells. We thus hypothesize that TOX expression is connected to phenotypic characteristics of dysfunctional NK cells rather than effector capacities. This is in accordance with the recent study from et al. Sekine et al., 2020 mentioned by the third reviewer.

**Author response image 4. respfig4:** Representative FACS-plot showing TOX and T-BET coexpression in total NK cells of a CHB patient.

5) To address their hypothesis on NFAT involvement in NK cell exhaustion and TOX expression the authors stimulate NK cells in vitro with ionomycin and show that pre-treatment with ionomycin renders NK cells hyporesponsive. They titrate the effect of ionomycin and find an ionomycin concentration which is inducing a reduction of IFNγ response without affecting degranulation. While the reduction of IFNγ response in this experiment is observed as in chronic HBV infection, this model should be validated before making any claims.

Following the reviewer’s recommendations, we carefully characterized this model. We first verified that ionomycin indeed induced NFAT1 nuclear translocation in NK cells after a 1hour treatment (Figure 6B). This resulted in increased TOX levels as soon as 4hours after the beginning of the treatment (Figure 6C). TOX and LAG3 levels correlated after an overnight treatment with ionomycin further underlying the convergence of this model with the physiological situation in CHB patient (Figure 6D). Regarding the functional parameters, we confirmed in a separate experiment on 5 HD that the ionomycin treatment indeed renders NK cells dysfunctional (Figure 6E). The dichotomy between IFN-γ and degranulation capacity is however less striking than in our pilot experiments. However, we still observed that IFN-γ is lost after treating the cells with 100nM ionomycin while degranulation is partially conserved.

It is not clear how treating with a higher concentration of ionomycin can mimic NK cell exhaustion that occurs over months or years. Theoretically, it cannot be a transient over-flux of calcium that initiates the expression of TOX and leading to NK cell exhaustion.

We agree with the reviewer that ionomycin treatment is a coarse model of the process leading to NK cell dysfunction in CHB patients. However, if we agree with the reviewer that our model does not take the “time factor” into account, we would respectfully disagree on the time-scale involved to induce dysfunction in vivo. It is true that patients have been infected for years with HBV, however, it is unlikely that the majority of the circulating NK cell population has such an extended half-life. Our hypothesis is that NK cells are constantly rendered dysfunctional on a time scale of days to weeks. Mechanistically, the fact that a transcriptional signature of “partnerless-NFAT” is detected shows that this factor is chronically activated. Whether this chronic activation involves chronic calcium flux or another unknown pathway is currently unclear. Understanding the molecular mechanism at play in vivo leading to an unbalanced activation of the NFAT pathway is still a matter of debate in the T cell field and probably involves defective co-stimulation. In the NK cell field we are limited to hypothesis as stated in the Discussion.

NFAT/Calcineurin could play a role in the formation of NK cell exhaustion.

Indeed, our working hypothesis is that calcium flux activates the Calcineurin/NFAT axis which in turn promotes dysfunction via various effectors. We made this hypothesis clearer in the revised version of the manuscript.

However, the over-activation of NK cells from healthy control does not prove that this mechanism is the cause of the pathological outcome.

We agree with the reviewer that the in vitro treatment with ionomycin can not on its own prove that chronic activation of this pathway is the cause of the dysfunctional phenotype. However, this result has to be linked with the fact that our RNAseq analysis detected a transcriptional signature consistent with chronic activation of “partnerless” NFAT. This obviously is not a proof but still a strong indication that this pathway participates in the dysfunctional phenotype.

Also, cells from patients might be investigated. Also, besides the ionomycin treatment performed, calcium flux experiment in sorted cells based on the phenotypes described would have been elegant.

We agree that investigating patient cells to directly demonstrate chronic NFAT activation would be ideal. In this respect, we quantified intra-nuclear NFAT1 in NK cells of CHB patients compared to HD using imaging cytometry. However, we did not detect any difference at basal state, nor after stimulation (data not shown). This might be due to the fact that cues of activation of the calcium-calcineurin-NFAT axis in patients in vivo are too elusive to be consistently detected ex vivo on frozen cells. In addition, NFAT activity might be necessary at the onset of the dysfunctional state but dispensable thereafter. Regarding calcium flux experiments, we were unable to perform them on sorted cells in preliminary experiments. Since these experiments promised to be technically very challenging on patient samples that require A3 safety level for their manipulation (French regulation) and given the fact that it was unclear how the results we would obtain would relate to chronic activation of the NFAT pathway, we did not perform them.

6) Given significant differences between the published characteristics of T cell exhaustion and findings described in the current work, it is not appropriate to conclude that the results are similar. This applies to both phenotypic and functional changes. For example, in multiple viral infection models, the decrease in IFN-γ production occurs in a step-wise manner during the progress of T cell exhaustion. In the current work, the authors show a significant and complete reduction of IFN-γ production in all the patients analyzed. Importantly, the number of T cells that produce multiple cytokines such as IFN-γ and TNF-α are reduced. However, it does not appear that these two cytokines are concurrently reduced in Hep-B patients. Another difference is that the NK cells from Hep-B patients are able to mediate normal cytotoxicity against K562 cells while the exhausted T cells are impaired in mediating this effector function. While it may be true that the NK cells in the Hep-B patients undergoing exhaustion, it may not be appropriate to equate this phenomenon with that seen in T cells.

We agree with the reviewer that both phenomena: T cell exhaustion and NK cell dysfunction in CHB patients cannot be completely equated at the functional level. This is the reason why we kept the term “dysfunctional” to describe the NK cell state in CHB patients. We also made several changes in the manuscript (including the Title) to clarify that both phenomena were not identical but rather shared some important features. However, we would like to argue that the data presented here indicate that T cell exhaustion and NK cell dysfunction are mechanistically comparable and rely on similar transcriptional mechanisms: unbalanced signal transduction downstream ITAM-receptors leading to preferential calcium pathway activation and chronic “partnerless NFAT” activity. The functional differences are secondary and probably related to ontogenetic differences between both cell types. In particular, the sensitivity of the different effector functions follows a different order in both cell type. Indeed, CD8 T cells lose first their capacity to secrete IL-2, a function NK cells do not possess, followed by IFN-γ/TNF-α secretion and cytotoxicity. In this respect, we would like to attract reviewers’ attention on the fact that high doses of ionomycin result in progressive loss of all effector functions including degranulation. This suggests that a step-wise process can also take place in NK cells and result in complete loss of effector functions. Whether this takes place in CHB patients upon stronger stimulation remains to be tested.

Finally, regarding TNF-α, we respectfully disagree with the reviewer. Indeed, we observed that decreased IFN-γ production correlated with decreased TNF-α production (See Figure 1—figure supplement 3A). Suggesting that both functions are affected by CHB, a situation very similar to the one observed in exhausted T cells.

7) The link that the authors are providing between mTOR-S6-NK cell exhaustion is not clear. The reduction in the phosphorylation of AKT is significant; but, moderate. Does this physiologically relevant? Does the alternate pathway mediated by PIM kinases is the one primarily affected in the NK cells from the Heo-B patients?

Our data in mouse models (Marçais et al., 2017) and in human (this study, Figure 3—figure supplement 1D) show that mTORC1 activity measured by pS6 is tightly linked to NK cell responsiveness. We thus think that the decrease in pS6 level is the most physiologically relevant one with respect to effector functions. The role of AKT is unclear at the moment. Despite the fact that we consistently find its activity decreased in hyporesponsive NK cells in mouse models (Marçais et al., 2017), basal pAKT levels did not correlate with functional parameters in the patient cohort studied in this manuscript. Regarding the alternate pathway involving PIM kinases we currently lack tools to reliably assess their role in CHB patients NK cells. We feel developing these tools is beyond the scope of this study.

8) Apart from NFAT, T-bet, BATF, EOMES, FOXO1, BLIMP1, and IRF4 have been implicated in playing a significant role in causing T cell exhaustion. What are the reasons that the gene signatures representing these transcription factors did not come through from the RNA sequencing analyses?

We agree with the reviewer that the transcriptional regulation of T cell exhaustion is indeed very complex and obviously not limited to NFAT, despite its proposed initiating role. Testing the role of other transcription factors (TF) in the induction of dysfunction in NK cells of CHB patients is indeed a relevant topic. However, transcriptional signatures are only defined for a few TF, in particular in a context of exhaustion. This is the primary technical reason that prevented us from detecting these signatures and still prevents us from looking specifically for them. Regarding T-BET, we co-stained for this transcription factor and TOX in an attempt to decipher their relationship. However, we could not detect any correlation (positive or negative) between these transcription factors.

In addition, a large part of the manuscript relates to TOX and its involvement in exhaustion. However, a recent article (Sekine et al., 2020) demonstrated that TOX is expressed by most circulating effector memory CD8^+^ T cell subsets and not exclusively linked to exhaustion. This is an important piece of work where such data might be integrated and invite to reinterpret results and conclusions.

We thank the reviewer for attracting our attention on this recently published article. Here again, it is clear that the importance of a TF has to be thought of in a particular context and that the same TF can be expressed in a variety of different situations. We now discuss the results of this study in our manuscript.

9) Figure 2 – as it seems, results display total NK cells which makes sometimes differences difficult to interpret, if possible, please provide in the supplement at least phenotype of Bright vs DIM NKG2A+ vs DIM NKG2A-.

Indeed, the results presented display total NK cells, unless otherwise stated. We agree with the reviewer that differentiation bias could impact the results. The fact that the Bright/Dim ratio is not affected in CHB patients argues against such a bias. To definitely settle this issue, we now provide as supplement the phenotypic data distinguishing Bright and Dim NK cells (Figure 2—figure supplement 1). It appears that most of the changes are more pronounced in the Dim subset at basal state. This is in accordance with the fact that TOX expression is principally affected in Dim cells as mentioned earlier. Regarding NKG2A status, as this staining was not systematically included in our panels, we can only provide the data for the following parameters: Perforin, TIGIT, NKp46 and DNAM-1 (See Author response image 5).

**Author response image 5. respfig5:** Mean fluorescence intensity (MFI) of the indicated markers was determined by flow cytometry on CD56Bright, CD56Dim NKG2A+ and CD56Dim NKG2A- NK cells among PBMCs from HD or CHB patients. Box-plots as well as values for each individual are represented.

10) Phosflow as well as mitochondrial analysis are always difficult to perform due to technical specificities, efficient detection of epitopes, atypical fluorescence leakages, or analysis of small shift differences. For both techniques, in order to highlight the quality of the datasets, please provide representative histograms as well as positive and negative controls, and gating strategy to further convince the readers.

We agree with the reviewer that these readouts are technically challenging. As requested, we provided representative data in the form of FACS plots (histograms and kinetic traces) and a gating strategy in the revised version of the manuscript. Regarding positive and negative controls, we felt that presenting them would complexify the result section. However, we performed the following controls, depicted in Author response image 6:

**Author response image 6. respfig6:** (A) Histogram overlays showing the level of the indicated phosphoepitope following 1h IL-15 stimulation in the presence or absence of the mTOR inhibitors Rapamycine (left) or Torin2 (right). (B) Kinetic of Kynurenine uptake with or without excess Leucine or Lysine, inhibitors of Slc7a6 or Slc7a5 respectively. (C) Histogram overlays showing the level of mitochondrial ROS with or without Antimycine A (complex III inhibitor) or Rotenone (complex I inhibitor).

- In NK cells, IL-15 stimulation is the best positive control for pS6, pAKT and pSTAT5 induction. Phosphorylation of S6 is inhibited by Rapamycin or Torin2 treatment while phosphorylation of AKT on Ser473 is inhibited by Torin2 only and phosphorylation of STAT5 is not affected by these inhibitors.

- Leucine excess inhibits kynurenine uptake via Slc7a5 while Lysine does not impact the uptake (Sinclair et al., 2018).

- For mitochondrial ROS detection, inhibition of the electron transport chain complex III with Antimycin A increases their production while inhibition of complex I with Rotenone decreases it.